# Insights from full-text analyses of the *Journal of the American Medical Association* and the *New England Journal of Medicine*

**Moustafa Abdalla[1,2]\*, Mohamed Abdalla[3,4], Salwa Abdalla[3], Mohamed Saad[5], David S Jones[1,6], Scott H Podolsky[1]\***

[1]Harvard Medical School, Boston, United States; [2]Department of Statistics, University of Oxford, Oxford, United Kingdom; [3]Department of Computer Science, University of Toronto, Toronto, Canada; [4]The Vector Institute for Artificial Intelligence, Toronto, Canada; [5]University of Bahrain & the Royal Academy, Manama, Bahrain; [6]Department of the History of Science, Harvard University, Cambridge, United States

**\*For correspondence:**
moustafa_abdalla@hms.harvard.
edu (MA);
scott_podolsky@hms.harvard.
edu (SHP)

**Competing interest:** The authors declare that no competing interests exist.

**Abstract** Analysis of the content of medical journals enables us to frame the shifting scientific, material, ethical, and epistemic underpinnings of medicine over time, including today. Leveraging a dataset comprised of nearly half-a-million articles published in the *Journal of the American Medical Association* (*JAMA*) and the *New England Journal of Medicine* (*NEJM*) over the past 200 years, we (a) highlight the evolution of medical language, and its manifestations in shifts of usage and meaning, (b) examine traces of the medical profession's changing self-identity over time, reflected in its shifting ethical and epistemic underpinnings, (c) analyze medicine's material underpinnings and how we describe where medicine is practiced, (d) demonstrate how the occurrence of specific disease terms within the journals reflects the changing burden of disease itself over time and the interests and perspectives of authors and editors, and (e) showcase how this dataset can allow us to explore the evolution of modern medical ideas and further our understanding of how modern disease concepts came to be, and of the retained legacies of prior embedded values.

## Editor's evaluation

This work analyzed more than half a million peer-reviewed articles published in two high-impact medical journals. It provides insights into the evolution of medical practice, language, and values over the past two centuries. Thus, it helps us contextualize our understanding of change in medicine and medical beliefs over time.

## Introduction

Medicine and medical language evolve over time, and this evolution can manifest in thematic expansion and contraction of vocabulary, lexical, and semantic changes, and cultural shifts of usage and meaning. The *Journal of the American Medical Association* (*JAMA*, founded in 1883) and the *New England Journal of Medicine* (*NEJM*, founded in 1812) have played critical roles in the development of medical knowledge and practice. They serve as rich data sources for the exploration of such trends. It is possible to do basic searches through *JAMA*'s and *NEJM*'s search interfaces, but it is difficult to analyze trends over time or perform more sophisticated analyses. Other online databases, notably

PubMed and Web of Science, allow slightly more sophisticated searches, but neither realizes the potential of or even captures the full 137-year history of *JAMA* and the 209-year history of *NEJM*.

Over the past decade, progress in digital history, scientometrics, and computational linguistics has led to new approaches and techniques for the analysis of the 'big data' of medical publications (*Boyack et al., 2005*; *Börner, 2010*; *Jones et al., 2011*; *Weisz et al., 2017*; *Thompson et al., 2016*; *Westergaard et al., 2018*). In this manuscript, leveraging a dataset of nearly half-a-million articles published in *JAMA* and *NEJM* over the past 200 years, we demonstrate the kinds of analyses that are now possible and that can provide valuable insight into the history of medical knowledge, practice, values, and institutions.

## Results and discussion
### Constructing the dataset

To enable this computational analysis, we constructed a database of nearly all articles ever published in *JAMA* and *NEJM*. In this study, an article is defined as any document with a digital object identifier (DOI), a system that assigns a unique identifier to academic publications. Our *JAMA* database captured 278,461 articles published from 1883 to 2018, representing >91% of all articles ever published there. Similarly, *NEJM* DOIs and associated authorship metadata were downloaded from Crossref on May 1, 2020. Based on DOI counts, our *NEJM* database captured 182,675 articles published from 1812 to 2020, which represents >99.5% of all articles ever published in *NEJM*. The total dataset analyzed in this study was comprised of 461,136 unique articles. For article PDFs that have not been digitized to machine-encoded text, we performed optical character recognition (OCR) using Tesseract v4.1.1, an open-source OCR engine using both legacy and LSTM engines and default automatic page segmentation. No pre-processing was performed prior to OCR. Data curation and methodology is described in further detail in Materials and methods.

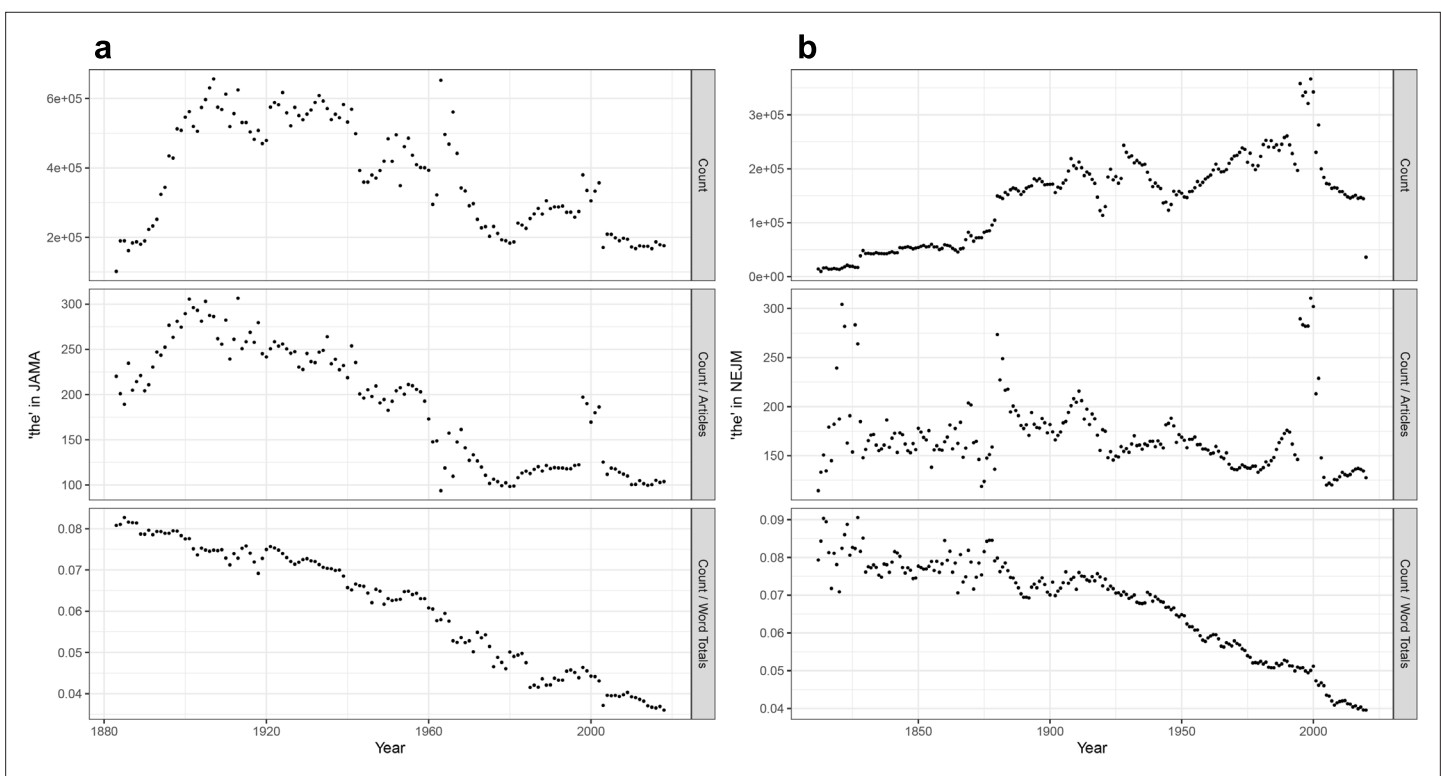

**Figure 1.** Time series analyses for the word 'the'.
(a) Frequency of the word 'the' in *Journal of the American Medical Association* (*JAMA*), with annual total raw counts (*top panel*), scaled by article counts (*middle panel*), and scaled by total number of words (*bottom panel*). (b) Frequency of the word 'the' in *New England Journal of Medicine* (*NEJM*), with annual total raw counts (*top panel*), scaled by article counts (*middle panel*), and scaled by total number of words (*bottom panel*).

## Time series

Perhaps the most elemental form of data-mining of publication datasets is the time series analysis of the changing prevalence of a term. Researchers often show a plot of increasing occurrences of a term over time to buttress arguments about the changing importance of a topic in the literature (*Tribble and Jones, 1997*; *Baron et al., 2009*; *Podolsky, 2015*; *Mane and Börner, 2004*). This can be done with the raw count of word usage, for instance in a dataset like PubMed. But there is a basic flaw to that methodology: the medical journal corpus has continuously expanded, such that the count of many words in that dataset will increase over time. This can be seen, for example, with a plot of occurrences of 'the' over time. An absolute time series within *NEJM* shows an increase over time, correlating with expansions in size of the journal (this is less apparent in *JAMA*). However, the new dataset allows calculation of a proper frequency analysis (occurrences of the target word in a given year divided by the total number of words published that year; *Figure 1*). This shows a marked decline in the use of 'the' over the 20th century in both journals, a phenomenon in the general literature well known to computational linguists (*Liberman, 2016*). This technique can then be used to look at words of greater medical meaning.

Some of the results confirm or lend nuance to our expectations. The occurrence of specific disease terms within *JAMA* and *NEJM* (*Figure 2a and b*) reflects the changing burden of disease itself over time – at least within the United States – and reflecting the interests and perspectives of authors and editors. Witness the 20th-century rise of 'cancer' or 'cardiac', or the 21st-century surge of 'coronavirus'. Tuberculosis and AIDS offer perhaps more instructive examples, showing a marked rise and fall over decades that reflects both their impact within the United States and the attention of researchers. However, this is also indicative of the relative invisibility of these two conditions, as they persist on a global scale long after the development of 'magic bullets' that promised to control both infections (*McMillen, 2015*; *HIV/AIDS JUNPo, 2018*; *Keshavjee and Farmer, 2012*).

We can similarly examine traces of the medical profession's changing self-identity over time, reflected in its shifting ethical and epistemic underpinnings. The emergence of concerns about ethics (and the rise of the field of bioethics) can be seen in the rise of words like 'ethical', 'autonomy', or 'consent' (*Figure 2*; *Rothman, 1991*; *Truog, 2012*). Concern with 'ethical', however, has diminished markedly since 1995, a signal worth further investigation. Even more strikingly, we see (*Figure 2e and f*) the rise of the clinical trial infrastructure and the epistemic grounding of medical science within such trials (*Bothwell et al., 2016*), reflected in the rise of 'controlled' and 'randomized', and still more dramatically, of 'trial' itself (with a corresponding decline in 'experiment').

When we turn to medicine's material underpinnings, an unexpected signal has emerged (*Figure 2g and h*). No historian of American medicine would be surprised by the increasing focus on the hospital – as site of care, education, and research – between 1850 and 1950 (*Rosenberg, 1987*; *Stevens, 1989*). But the dramatic decline in usage of the term, post-1950, is a surprising finding, given the presumably persisting centrality of the hospital in each of these domains. What accounts for this? Did the passage of the Hill-Burton Act in 1946, which led to the establishment of some 6800 new hospitals (*Health Resources and Services Administration, 1992*), establish the role of hospitals in health care such that they no longer needed to be discussed and debated? Or, conversely, does the decline reflect a shift in focus of medical care and education to the outpatient setting (*Figure 2g and h*)? The term 'clinic', however, nearly matches the rise and fall of hospital, with its peak in the 1940s. We welcome other hypotheses from readers.

Countless searches can be done for other terms believed to be of significance, and many of these will reveal unexpected signals that demand analysis and require more thorough historical investigation.

## Time series for bigrams and trigrams (sequential pairs and triplets of words)

Analogous searches can be conducted for bigrams, compared to the denominator of all bigrams within a particular corpus (and they can be similarly conducted for still longer combinations of words). For example, this sentence contains six bigrams. As with single words, some of these examples confirm or lend nuance to expectations, methodologically or conceptually. Methodologically, it may be noted that one of us had previously manually traced the rise and fall of the term 'personal equation' in the medical literature from 1850 to 1950 as a complex, multi-faceted term at times signifying individual patient or clinician variability, at other times signifying observer bias. The rise and fall of 'personal

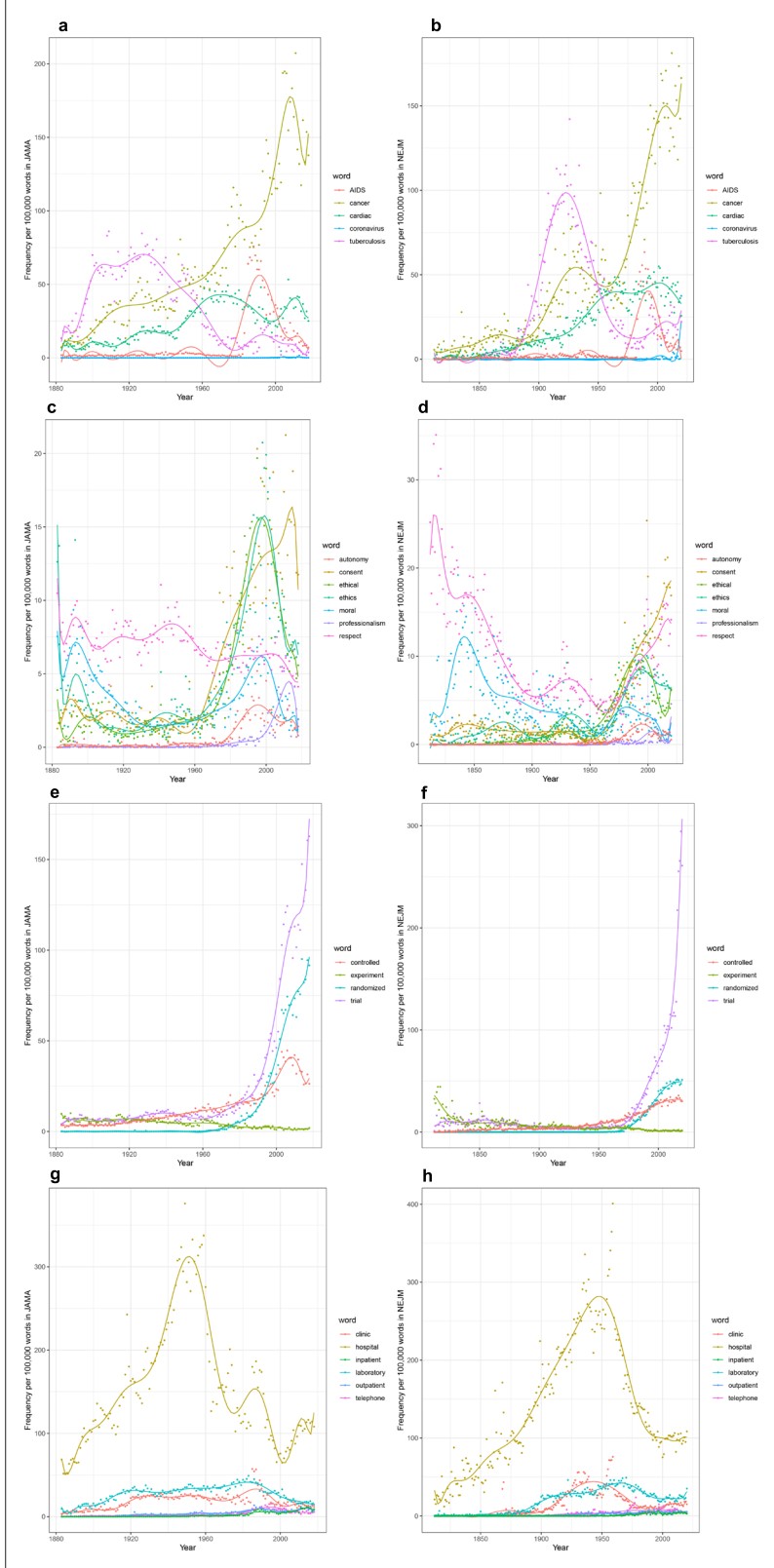

**Figure 2.** Time series plots for selected words, with frequency per 100,000 words as a function of year.
Time series plots, with frequency per 100,000 words as a function of year for: the words 'AIDS', 'cancer', 'cardiac',
'coronavirus', and 'tuberculosis' in the (**a**) *Journal of the American Medical Association* (*JAMA*) corpus (1883–2018)
and (**b**) *New England Journal of Medicine* (*NEJM*) corpus (1812–2020); the words 'autonomy', 'consent', 'ethical',

*Figure 2 continued on next page*

*Figure 2 continued*

'ethics', 'moral', 'professionalism', and 'respect' in (**c**) *JAMA* and (**d**) *NEJM*; for the words 'controlled', 'experiment', 'randomized', and 'trial' in (**e**) *JAMA* and (**f**) *NEJM*; and for the words 'clinic', 'hospital', 'inpatient', 'laboratory', 'outpatient', and 'telephone' in (**g**) *JAMA* and (**h**) *NEJM*.

equation' through the present bigram computation (*Figure 3a*) conforms well to the *NEJM* analysis conducted previously through examination of each instance of the term recognized by the *NEJM*'s own full-text search tool (*Brinkmann et al., 2019*).

Turning to novel searches, we see that 'antibiotic resistance' rises with the widespread recognition of resistant bacteria in the 1950s and of their horizontal transmission of resistance in the 1960s (*Figure 3b and c*). It appears to drop off in salience by the early 1980s, before a second wave of attention from the late 1980s onward as it became linked to emerging infections more broadly, and concerns about the capacity of the pharmaceutical industry to keep up with such newly resistant bacteria in particular (*Podolsky, 2015*; *Neu, 1992*; *Overton et al., 2021*). The late 20th-century rise of 'informed consent' (apparently comprising half of the uses of 'consent'; *Figure 3d and e*), fall of 'mental retardation', (*Figure 3f and g*), and swift rise and fall of 'managed care' (while 'health policy' rose and remained high; *Figure 3h and i*) again point to the ethical, semantic/conceptual, and material/infrastructural underpinnings of organized medicine.

Yet, there are again perhaps more instructive signals. Both 'breast cancer' and 'lung cancer' rise throughout the latter half of the 20th century (*Figure 3j and k*). However, despite the predominance in recent decades of lung cancer mortality over breast cancer mortality, 'breast cancer' is consistently mentioned at a higher rate (roughly twice the rate) than is 'lung cancer'. This parallels discrepancies seen in the ratio of National Institutes of Health funding to the burden of disease contributed by these two forms of cancer (*Gross et al., 1999*). Does this represent differential stigmatization of lung cancer versus breast cancer patients? The mobilization of attention, advocacy, and funding around one versus the other condition? Again, such signals warrant additional investigation.

'Myocardial infarction', like cancer, rises dramatically over the 20th century (*Figure 3l and m*). But here the trajectories in the two journals are distinct. While the plot of *NEJM* data shows a steady rise to its peak in 2000, the plot of *JAMA* data shows a bimodal distribution with one peak in the 1970s, followed by a notable dip, and then a rise to a second peak in the early 2000s. Similar trajectories are seen with 'heart attack' in the two journals (though at much lower prevalence). It is hard to guess what might have led to the decrease in prevalence of myocardial infarction in *JAMA* at a time when it remained the leading cause of death. Did this reflect a shift in editorial policy? Was a different term briefly favored? Again, further investigation would be required.

To enable further research to distill these trends (both single-word frequencies and higher-order ngrams), as well as to support other interrogations for unexpected signals and stories, we provide an ngram site to enable users to visualize and model longitudinal trends in *NEJM* and *JAMA* (*Figure 4*) at: https://countway.harvard.edu/center-history-medicine/center-services/looking-glass.

## Word meanings

One challenge of the time series analyses is that the meaning and usage of a specific term can change over time. Consider the case of 'patent', which shows two peaks, one in the 1840s and another in the early 2000s (*Figure 5*). The new dataset can be analyzed to explore this. The technique of vector representations (referred to as 'word embeddings' among computational linguists) can be used to identify the other words most closely associated with the target term in a specific time period. Fundamentally, these techniques seek to define a numerical address (a vector) for every word based on the company it keeps: more co-occurrences translate to two words inhabiting a similar neighborhood in an abstract n-dimensional space. The numerical addresses for most words (e.g., 'the') are fixed and do not change, while others vary and evolve over time (see Materials and methods). Such an analysis answers the puzzle for 'patent' (*Tables 1 and 2*; *Supplementary files 1 and 2*): the two peaks of occurrences reflect distinct sets of meaning, with patent medicines explaining the first peak (e.g., 'patent' occurs in association with nostrum, proprietary, fakish, secret, exploiter, evil, dopers, and quackery) (*Gabriel, 2014*; *Young, 1987*), while a mix of inventor's patents (e.g., inventors, coinventors, device) and congenital heart disease (patent foramen ovale, patent ductus arteriosus) explains the second peak.

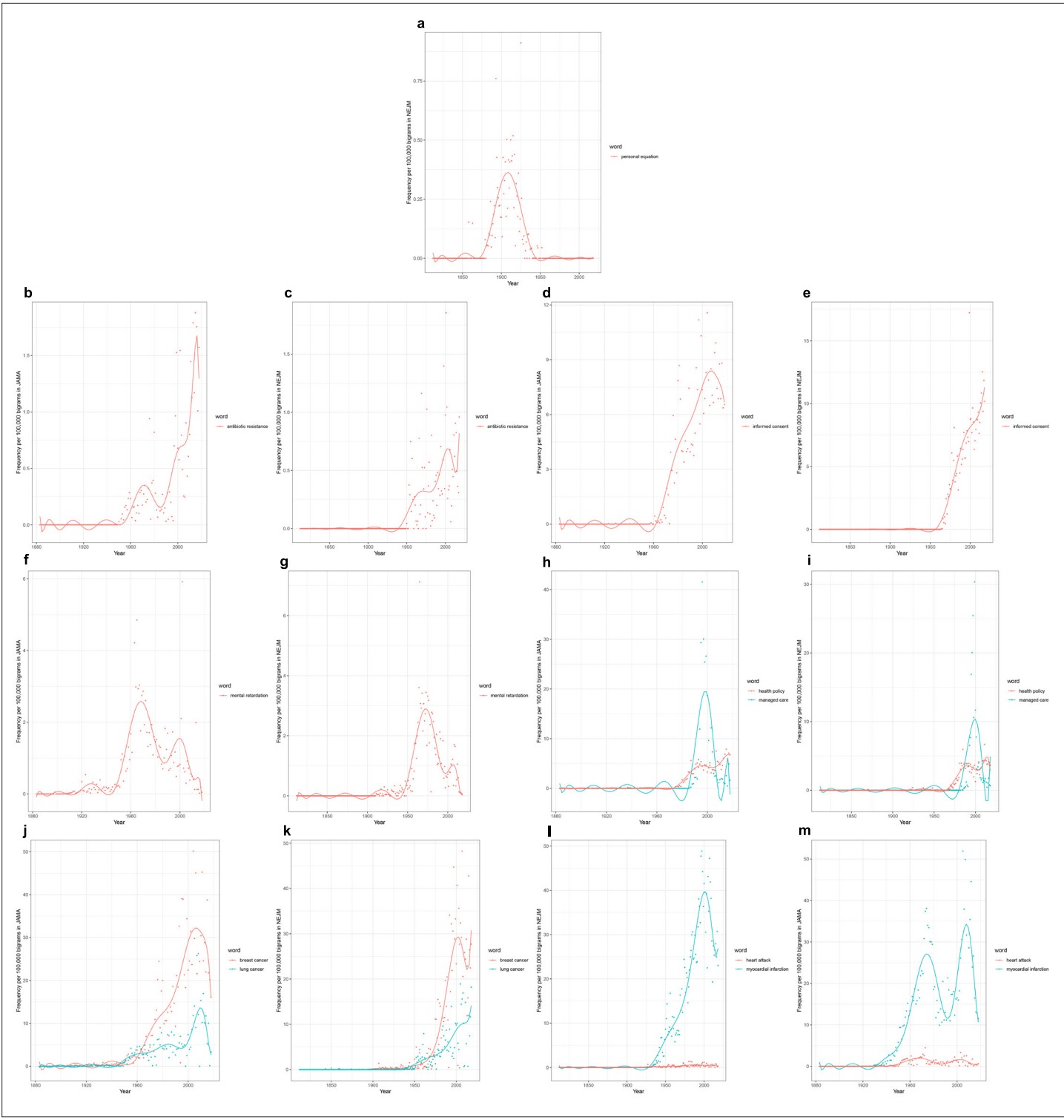

**Figure 3.** Time series plots for selected bigrams, with frequency per 100,000 bigrams as a function of year.
Time series plots, with frequency per 100,000 bigrams as a function of year, for: the bigram 'personal equation' in the (**a**) *New England Journal of Medicine* (*NEJM*) corpus; the bigram 'antibiotic resistance' in the (**b**) *Journal of the American Medical Association* (*JAMA*) corpus (1883–2018), and (**c**) *NEJM* corpus (1812–2020); the bigram 'informed consent' in (**d**) *JAMA* and (**e**) *NEJM*; the bigram 'mental retardation' in (**f**) *JAMA* and (**g**) *NEJM*; the bigrams 'health policy' and 'managed care' in (**h**) *JAMA* and (**i**) *NEJM*; the bigrams 'breast cancer' and 'lung cancer' in (**j**) *JAMA* and (**k**) *NEJM*; and the bigrams 'myocardial infarction' and 'heart attack' in (**l**) *JAMA* and (**m**) *NEJM*.

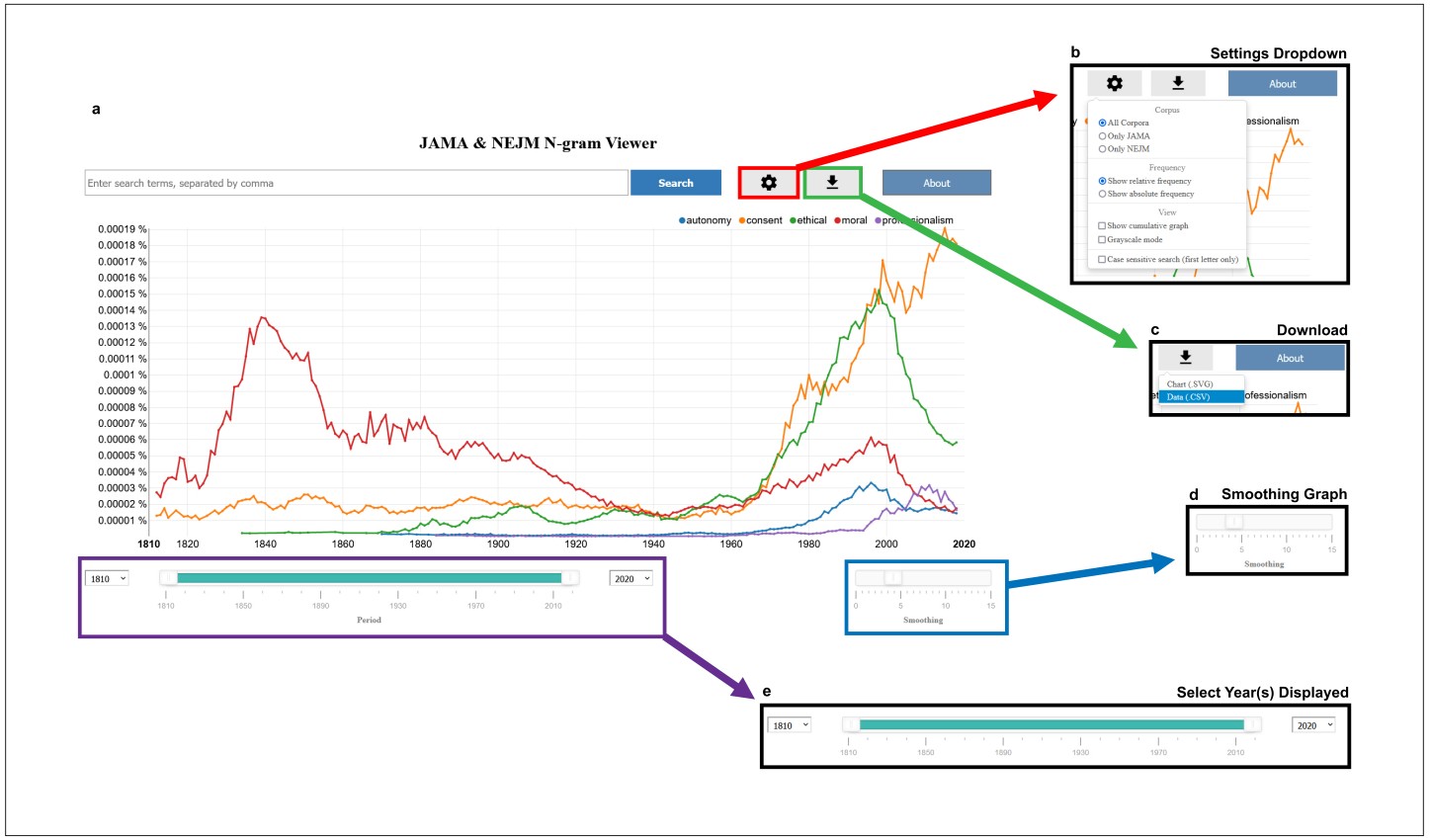

**Figure 4.** Screenshots of our *Journal of the American Medical Association* (*JAMA*) and *New England Journal of Medicine (NEJM)* ngram viewer. (**a**) The landing page of our website defaults to the word frequencies highlighted in *Figure 3*. (**b**) Analysis functionality includes calculating both the relative and absolute frequencies in both corpora, *NEJM* only, and *JAMA* only, as well as plotting cumulative frequencies, changing color of lines to grayscale, and enabling case-sensitive searches. (**c**) Users can download both the plot as a scalable vector graphics (svg) file and the raw data for their query as a csv file. (**d**) Frequency lines can be smoothed with the 'smoothing bar' on the bottom right-hand corner of the screen. (**e**) Time series plots can also be limited to a particular window/time period by changing the time bar on the bottom of the screen with the drag-and-drop timeline or with the corresponding dropdown menus.

Many words are still more informative. We next focus on four charged ones: abortion, bias, defective, and race.

Abortion (*Tables 1 and 2*; *Supplementary files 3 and 4*) has been an enduring problem for the medical profession in the United States, with the procedure widely tolerated in the early 19th century, criminalized in the late 19th century, decriminalized in the 1970s, and contested ever since (*Mohr, 1979*; *Reagan, 1997*; *De Ville, 1992*). Early associations include causes (e.g., smallpox) and related processes (e.g., miscarriage, pregnancy, menorrhagia, labor). 'Criminal' first appears in *NEJM* in the 1850s, corresponding with the medical campaign to ban the procedure, and remains atop the list through the early 1900s. Contraception first appears in the 1930s, but does not become a strong association until the 1950s in *JAMA* and the 1960s in *NEJM*. Legalization first appears in the 1970s and becomes more prominent by the 1990s. Specific articles would need to be examined to determine whether this reflects an emerging argument that abortion could be justified in the case of contraceptive failure, or (in the 1960s) part of a broader conversation about the emerging 'right to privacy' that was the key to the Supreme Court decisions that legalized first contraception (Griswold v. Connecticut) and then abortion (Roe v. Wade) (*Mohr, 1979*; *Reagan, 1997*; *Ziegler, 2020*). In *NEJM*, religion and religious also first appear in the 1970s, and only break into the top 10 (religious) in the 2010s, reflecting an important observation made by historians that the abortion debate in the United States was not couched in religious terms until quite recently. However, both religion and religious are strikingly less prominent among *JAMA*'s nearest terms – neither word falls within the top 30 associations at any point. This is a notable finding, possibly reflecting editorial differences and

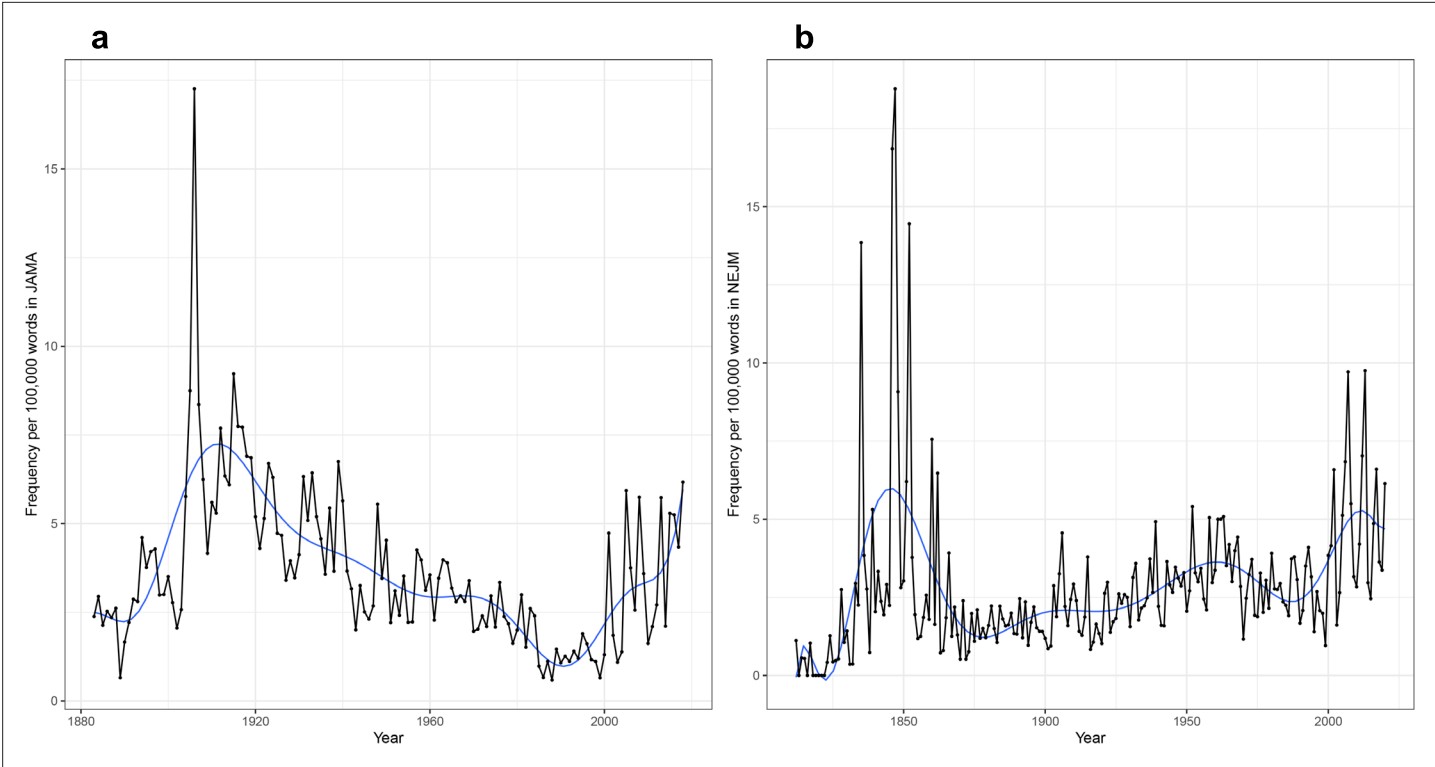

**Figure 5.** Time series plot, with frequency per 100,000 words as a function of year, for the word 'patent' in the (**a**) *Journal of the American Medical Association* (*JAMA*) corpus and the (**b**) *New England Journal of Medicine* (*NEJM*) corpus.

suggests that there may be dissimilarities in the academic discussion regarding abortion between the two journals.

Bias rose precipitously in usage, in both *JAMA* and *NEJM*, in the latter half of the 20th century (though falling, unexpectedly, in the 21st century in *NEJM*; *Figure 6*). As of the 1920s, its top 20 associations are moral and vernacular (*Tables 1 and 2*; *Supplementary files 5 and 6*). In *JAMA*, bias is associated with words such as prejudice, insincerity, deluding, untruthfulness, and wrongness. Similarly, in *NEJM*, the top 20 most-related terms for bias include deception, guilt, insincerity, malevolence, sinfulness, disgust, and pettiness. But we already see hints of later, less-conscious misperception, with misjudgment, denial, and subconscious included (as was 'sham') in the 1920s; and by the 1950s, 'uncertainty' is the #3 related term in *JAMA* and 'subjectivity' has become the #4 related term in *NEJM*. To understand the meaning of these moral meanings of bias from the 1920s to 1960s, it would be necessary to examine specific articles in detail. Is bias a problem in physicians or patients? In clinicians or researchers? Tools like ours can suggest puzzling findings, but traditional modes of historical analysis are required to understand them. By the 1970s, bias has largely become a statistical clinical trial term – with misclassification, selection, retrospective, errors, confounding, and validity all in the top 20 – but its moral associations are still apparent in such associations further down the list as unverifiable, blatantly, recalcitrance, and antisocial, demonstrating medicine's linked moral and epistemic commitments.

Defective (*Tables 1 and 2*; *Supplementary files 7 and 8*), a word with strong moral valences, has diffuse (and hard to interpret) associations in the 19th century. A new set of associations rises to prominence in the early 20th century (present in both *JAMA* and *NEJM*) – impaired, congenitally, feebleminded, mentally, delinquents, retarded, idiot, and others – a reflection of the rise of the eugenics movement in the United States and its concern with supposedly defective people (*Kevles, 1995*; *Paul, 1998*). While eugenics has faded from medical discourse, 'defective' persists, now almost exclusively in association with genes (tlr1, cftr, nlrp3, dysregulated, dock2, repression, etc.). This reflects the emergence of modern genetics, a very different science of inter-individual variability than that which eugenics had promised. But genetics still encodes a language of value judgments, sharing eugenics'

**Table 1.** Table of the 10 most-related terms for the words 'patent', 'abortion', 'bias', 'defective', and 'race' in *Journal of the American Medical Association* (JAMA), stratified per decade.

For historical accuracy, we intentionally did not filter misspellings and/or offensive words.

**Patent**

| 1880–1889 | 1890–1899 | 1900–1909 | 1910–1919 | 1920–1929 | 1930–1939 | 1940–1949 | 1950–1959 | 1960–1969 | 1970–1979 | 1980–1989 | 1990–1999 | 2000–2009 | 2010–2019 |
|---|---|---|---|---|---|---|---|---|---|---|---|---|---|
| proprietary | proprietary | proprietary | proprietary | trademark | histeen | arteriosus | ductus | ductus | ductus | ductus | ductus | coinventors | coinventors |
| fakish | venders | seeret | propritary | proprietary | exploiter | ductus | arteriosus | arteriosus | arteriosus | arteriosus | arteriosus | coinventor | royalties |
| readymade | exploiter | propritary | quack | exploiter | proprietary | ovale | coarctation | ovale | arteriosis | duetus | foramen | ovale | therapeutics |
| mostrums | putent | sceret | quackery | nurito | hard | proprietary | anomalous | arteriosis | bcv | arteriosis | databanks | foramen | biosimilar |
| polypharmaceutic | frandulent | mostrum | nostrum | citrophan | kolloyd | arteriosis | ovale | duetus | arteries | ovale | inventors | patents | coinventor |
| bootleggers | inventors | deception | seeret | patented | wheat | quackery | arteriosis | postductal | duetus | peritoneovenous | proprietary | biosimilar | extensions |
| proprie | proprie | exploiter | exploiters | distributer | trademark | duetus | tracheoesophageal | stenosis | grafts | bcv | defect | biosimilars | corevalve |
| fraudulence | rankest | faker | exploiter | benetol | distributer | medieines | interatrial | arteries | anastomotic | arteries | aortopulmonary | closure | sapien |
| abominations | unblushing | quack | venders | exploiters | nurito | stenosis | atresia | botallo | anastomosis | eustachian | deviees | ndas | equity |
| tised | seeret | proprietorship | fakers | purifico | exploiters | valves | tricuspid | bifurcating | coarctation | intraventricular | coinventor | nonexclusive | chimerix |

**Abortion**

| 1880–1889 | 1890–1899 | 1900–1909 | 1910–1919 | 1920–1929 | 1930–1939 | 1940–1949 | 1950–1959 | 1960–1969 | 1970–1979 | 1980–1989 | 1990–1999 | 2000–2009 | 2010–2019 |
|---|---|---|---|---|---|---|---|---|---|---|---|---|---|
| premature | miscarriage | miscarriage | delivery | miscarriage | miscarriage | miscarriage | miscarriage | pregnancy | abortions | abortions | contraception | abortions | miscarriage |
| miscarriage | previa | criminal | miscarriage | pregnancy | habitual | habitual | contraception | abortions | pregnancy | contraception | abortions | tellingly | stillbirth |
| ectopic | labor | pregnancy | pregnancy | puerperal | puerperal | ahortion | pregnancy | threatened | euthanasia | pregnancy | birth | eapen | birth |
| labor | endometritis | delivery | eclampsin | abortion | criminal | pregnancy | abortion | intrauterine | midtrimester | euthanasia | criminalized | offsite | obstetric |
| ufero | pregnancy | labor | tubal | delivery | ahortion | threatened | habitual | miscarriage | abor | ectopic | contraception | infertility | intrauterine |
| accouchment | ectopic | ectopic | inbor | pregnant | stillbirth | criminal | aborted | contraception | legal | midtrimester | midtrimester | legalization | spontaneous |
| delivery | puerperal | pravia | abortion | criminal | abortions | hydatidiform | stillbirth | infertility | amniocentesis | unwanted | secondtrimester | mischaracterizes | cdmr |
| faetus | eclampsia | eclampsia | puerperal | spontaneous | pregnancy | menorrhagia | delivery | labor | criminal | tubal | eugenic | pregnancy | pregnancy |
| hzmorrhage | delivery | premature | premature | eclampsia | threatened | menstruation | threatened | delivery | unwanted | fetus | sterilization | legalizing | nonlegal |
| criminal | intrauterine | previa | placenta | dystocia | delivery | ectopic | abortions | postpartum | sterilization | ovum | amniocentesis | contraception | preeclampsia |

**Bias**

| 1880–1889 | 1890–1899 | 1900–1909 | 1910–1919 | 1920–1929 | 1930–1939 | 1940–1949 | 1950–1959 | 1960–1969 | 1970–1979 | 1980–1989 | 1990–1999 | 2000–2009 | 2010–2019 |
|---|---|---|---|---|---|---|---|---|---|---|---|---|---|
| stigma | guilt | insincerity | condonation | prejudice | reasonings | prejudice | overgeneralization | biases | biases | biases | biases | biases | biases |
| beneficence | preferences | prepossession | reasonahle | insincerity | argumentation | disadvantageously | overmatching | prejudice | selection | biased | biased | biased | selection |
| vileness | inborn | instincts | adjudge | pretentions | triteness | didacticism | uncertainty | biased | overmatching | selection | selection | selection | confounding |
| jearning | denial | partisanship | insincerity | prepossession | marshalling | overload | foreknowledge | repetition | biased | misclassification | nondifferential | misclassification | unmeasured |
| responsibility | trammels | reasonings | eruel | partisanship | didacticism | preconception | prejudice | errors | selfselection | nondifferential | selfselection | confounding | biased |
| inflictions | jurymen | perverseness | premeditation | depreciating | fairness | selfcentered | nonpublication | overmatching | biasing | confounding | misclassification | unmeasured | misclassification |
| imbues | prepossession | disinterestedness | meddlesomeness | deluding | prejudice | expectable | oversimplifications | generalization | generalization | overmatching | confounding | nondifferential | attrition |

*Table 1 continued on next page*

*Table 1 continued*

**Bias**

| 1880–1889 | 1890–1899 | 1900–1909 | 1910–1919 | 1920–1929 | 1930–1939 | 1940–1949 | 1950–1959 | 1960–1969 | 1970–1979 | 1980–1989 | 1990–1999 | 2000–2009 | 2010–2019 |
|---|---|---|---|---|---|---|---|---|---|---|---|---|---|
| adherence | hypnotizer | diseredit | untruthfulness | unreserved | adducing | hewed | artificiality | validity | nonindependence | imprecision | masking | funnel | blinding |
| prostitution | misconstruction | eruel | jurymen | indiscriminating | doctrinaire | histrionics | generalization | discrimination | overgeneralization | selfselection | blinding | selfselection | validity |
| equity | validity | enthusinsm | tactless | unkindly | impressiveness | neutrally | interpretational | preferences | oversimplifications | underreporting | misreporting | imprecision | imprecision |

**Defective**

| 1880–1889 | 1890–1899 | 1900–1909 | 1910–1919 | 1920–1929 | 1930–1939 | 1940–1949 | 1950–1959 | 1960–1969 | 1970–1979 | 1980–1989 | 1990–1999 | 2000–2009 | 2010–2019 |
|---|---|---|---|---|---|---|---|---|---|---|---|---|---|
| calorification | impaired | impaired | defects | defects | mentally | impaired | impaired | impaired | chemotaxis | eukaryotic | mutant | mitochondria | tlr1 |
| deficiency | defects | imperfect | defect | impaired | defects | defects | defects | fibrinase | defect | defect | ribozyme | cftr | autophagy |
| malnutrition | malnutrition | defects | defeets | poor | poor | congenitally | retardation | abnormal | impaired | mitochondrial | fasl | mitochondrial | pvhl |
| abnormalities | imperfect | deficiency | poor | mentally | defect | abnormal | congenitally | defect | glycosphingolipid | progenitor | germline | autophagy | cardiomyocyte |
| nil | retardation | deficiencies | mentally | abnormal | congenitally | retardation | defect | recessively | suppressor | mutant | genetically | ryanodine | cd47 |
| imperfect | defect | abnormal | abnormalities | congenitally | cardiopathic | mentally | mentally | prtase | antiglomerular | clone | deregulated | pvhl | tlr2 |
| impairment | nutrition | defect | abnormal | defect | idiot | cardiopathic | abnormal | neurovisceral | fibrinase | retroviral | oncogene | misfolded | mfn2 |
| deciduous | congenitally | imperfect | impaired | cardiopathic | aberrations | unstable | impairment | congenitally | mitochondrial | genetically | phenotype | sod1 | hmga1 |
| emotional | impairment | mentally | insufficient | impairment | offspring | blind | poor | deficiency | leukotactic | gene | fibrillin | mutant | txnip |
| immobility | disturbances | anomalies | malnutrition | deafmutes | impaired | poor | deficiency | heterozygotic | neurofilament | chaperone | deletion | degradation | mapk |

**Race**

| 1880–1889 | 1890–1899 | 1900–1909 | 1910–1919 | 1920–1929 | 1930–1939 | 1940–1949 | 1950–1959 | 1960–1969 | 1970–1979 | 1980–1989 | 1990–1999 | 2000–2009 | 2010–2019 |
|---|---|---|---|---|---|---|---|---|---|---|---|---|---|
| happiness | offspring | negro | community | races | sex | sex | sex | sex | sex | sex | ethnicity | ethnicity | ethnicity |
| offspring | peoples | human | sex | native | races | nationality | ethnic | nationality | age | ethnic | sex | sex | sex |
| intellectual | happiness | ancestry | negro | peoples | ancestry | geographic | ancestry | age | marital | marital | gender | gender | hispanic |
| fetichism | african | mankind | social | ancestry | ethnic | creed | nationality | geographic | nativity | gender | marital | ethnic | age |
| religion | ancestry | hereditary | human | dominant | jews | nativity | nativity | marital | ethnic | ethnicity | status | status | marital |
| brutes | hybrid | peoples | moral | recessive | nationality | marital | negroid | religion | religion | age | age | selfidentified | status |
| populations | instinct | uncivilized | hereditary | sex | hereditary | negro | marital | birthdate | age | status | hispanic | hispanic | origin |
| tenderest | esquimaux | degeneracy | offspring | populations | intermarriage | happiness | mating | geography | smoking | hispanic | selfidentified | age | selfidentified |
| morals | religion | zoophilism | socinl | community | family | indians | temperament | habits | socioeconomic | characteristics | hispanie | sociodemographic | asian |
| gambling | mankind | races | eugenies | interbreeding | tabus | age | geographic | religion | distribution | socioeconomic | marital | marital | demographics |

Notes: All 200 most-related terms are available in Supplementary files.

**Table 2.** Table of the 10 most-related terms for the words 'patent', 'abortion', 'bias', 'defective', and 'race' in *New England Journal of Medicine (NEJM)*, stratified per decade.

Given space constraints and to facilitate comparison with **Table 1**, we limit the tables to post-1880s; full analyses (1810–2020) are provided in eTables. For historical accuracy, we intentionally did not filter misspellings and/or offensive words.

**Patent**

| 1880–1889 | 1890–1899 | 1900–1909 | 1910–1919 | 1920–1929 | 1930–1939 | 1940–1949 | 1950–1959 | 1960–1969 | 1970–1979 | 1980–1989 | 1990–1999 | 2000–2009 | 2010–2019 |
|---|---|---|---|---|---|---|---|---|---|---|---|---|---|
| compounder | revalenta | proprietary | proprietary | ductus | arteriosus | ductus | ductus | ductus | ductus | ductus | arteriosus | foramen | ovale |
| peddling | proprietary | quack | quack | arteriosus | ductus | arteriosus | arteriosus | arteriosus | arteriosus | arteriosus | ductus | ovale | foramen |
| revalenta | quack | revalenta | quackery | eomron | ovale | ovale | ovale | ovale | ovale | licensing | foramen | ductus | amplatzer |
| affixing | revalenta | wholesaler | eomron | magna | incompetence | transposition | tetralogy | foramen | saphenous | artery | ovale | coinventors | ductus |
| fouled | wholesaler | eomron | magna | narroav | arteriosas | septal | foramen | interatrial | arteries | cava | amplatzer | arteriosus | arteriosus |
| baited | venders | venders | core | arteriosas | truncus | bicuspid | fallot | truncus | grafts | pda | occluder | amplatzer | device |
| rummage | quack | blameworthy | thyroglossal | ovale | tricuspid | coarctation | arteries | dextrocardia | foramen | equity | infarctrelated | inventors | corevalve |
| vulcanized | hawkers | rummage | cluster | papilla | septal | interauricular | truncus | stenosis | stenosis | ventriculoamniotic | starflex | coinventor | coinventors |
| utensil | compounder | leak | appendage | unsafe | interauricular | dilated | toleft | stenosis | atresia | leftto | anomalous | patents | patents |
| venders | tÃ°pelo | affixing | medicinal | tight | urachus | arteriosas | dilated | toleft | stenosis | atresia | leftto | anomalous | venosus |
| venders | vending | shellfish | dangerof | evil | urachus | stenosis | interatrial | leftto | coarctation | amplatzer | venosus | pfo | coinventor |

**Abortion**

| 1880–1889 | 1890–1899 | 1900–1909 | 1910–1919 | 1920–1929 | 1930–1939 | 1940–1949 | 1950–1959 | 1960–1969 | 1970–1979 | 1980–1989 | 1990–1999 | 2000–2009 | 2010–2019 |
|---|---|---|---|---|---|---|---|---|---|---|---|---|---|
| premature | miscarriage | miscarriage | eetopic | ectopic | miscarriage | pregnancy | threatened | miscarriage | pregnancy | abortions | miscarriage | abortions | sterilization |
| miscarriage | criminal | tubal | justifiability | eclampsia | ectopic | septic | intrauterine | habitual | septic | miscarriage | abortions | miscarriage | contraception |
| delivery | sepsis | premature | eclampsia | tubal | eclampsia | stillbirth | miscarriage | stillbirth | contraception | stillbirth | contraception | sterilization | labor |
| threatened | premature | ectopic | tubal | eetopic | intrauterine | placentae | habitual | placentae | euthanasia | pregnancy | legalization | mifepristone | unwanted |
| eclampsia | puerperal | sepsis | divorce | septicemia | placentae | stillbirth | stillbirth | pregnancy | sterilization | infertility | pregnancy | pregnancy | refusal |
| subinvolution | pregnancy | delivery | postpartum | ablatio | previa | puerperal | puerperal | tubal | amniocentesis | firsttrimester | assisted | stillbirth | abortions |
| replacement | eclampsia | pregnancy | intrauterine | previa | pregnaney | toxemia | abortions | labor | fetus | aborted | sterilization | labor | marriage |
| pregnancy | tubal | criminal | threatened | accidental | accreta | previa | abruptio | contraception | midtrimester | contraception | mifepristone | legalization | religious |
| menorrhagia | endometritis | fetus | miscarriage | postpartum | ectopic | pregnancy | hyperemesis | fetus | unwanted | tubal | unwanted | bendectin | couples |
| pregnaucy | previa | previa | prsevia | toxemia | toxemia | tubal | praevia | abortions | stillbirth | fetus | legal | multifetal | conscience |

**Bias**

| 1880–1889 | 1890–1899 | 1900–1909 | 1910–1919 | 1920–1929 | 1930–1939 | 1940–1949 | 1950–1959 | 1960–1969 | 1970–1979 | 1980–1989 | 1990–1999 | 2000–2009 | 2010–2019 |
|---|---|---|---|---|---|---|---|---|---|---|---|---|---|
| denial | aggressive | insincerity | avowal | deception | misconstruction | misjudgment | nonexistence | prejudice | biases | misclassification | biases | misclassification | biases |
| juryman | prepossession | sinfulness | eritic | symbolic | fairminded | procrastination | prejudice | pervasive | misclassification | biased | misclassification | biases | misclassification |
| unwisdom | conscience | unwisdom | expositor | inborn | blinded | discrimination | provable | overenthusiasm | biased | misclassification | biased | confounding | selection |
| scoffing | fairness | carelessness | carnestly | guilt | verbal | pessimism | subjectivity | generalization | selection | biasing | selection | biased | biasing |
| conscience | repression | unmanly | conviction | conscience | broadminded | misconception | generalization | hunches | biasing | selection | confounding | selection | biased |
| prudish | denial | biassed | scoffing | insincerity | disavowal | grudges | denial | biases | prejudice | imprecision | nondifferential | nondifferential | confounding |
| jealousy | slovenliness | unsettle | witticism | subconscious | ecriticism | depreciatory | fantasy | undocumented | imprecision | nondifferential | overreporting | biasing | nondifferential |

*Table 2 continued on next page*

*Table 2 continued*

**Bias**

| 1880–1889 | 1890–1899 | 1900–1909 | 1910–1919 | 1920–1929 | 1930–1939 | 1940–1949 | 1950–1959 | 1960–1969 | 1970–1979 | 1980–1989 | 1990–1999 | 2000–2009 | 2010–2019 |
|---|---|---|---|---|---|---|---|---|---|---|---|---|---|
| excusing | forbearing | metaphysical | ecriticism | intolerant | misjudgment | gynic | denigration | subjectivity | retrospective | blinding | biasing | underpowered | nonresponse |
| impolitic | vindictiveness | deprecation | recreant | malevolence | dogmatize | shirks | incongruities | injustices | problem | representativeness | design | overreporting | attrition |
| impiety | disabused | dispossess | dispossess | sinfulness | indelicacy | comprehension | misjudgment | denial | nondifferential | generalization | ascertainment | nonresponse | blinding |

**Defective**

| 1880–1889 | 1890–1899 | 1900–1909 | 1910–1919 | 1920–1929 | 1930–1939 | 1940–1949 | 1950–1959 | 1960–1969 | 1970–1979 | 1980–1989 | 1990–1999 | 2000–2009 | 2010–2019 |
|---|---|---|---|---|---|---|---|---|---|---|---|---|---|
| defects | impaired | defects | mentally | defects | retardation | unbalance | impaired | impaired | impaired | defect | expression | dysregulated | nlrp3 |
| defect | plumbing | unstable | defects | mentally | defects | regulatory | germ | deficiency | defect | impaired | constitutive | repression | senescence |
| overcrowding | defect | defeetive | defect | malnutrition | impaired | maladjusted | accelerated | congenitally | deficiency | pretranslational | alas2 | nlrp3 | dysregulated |
| plumbing | defeetive | defect | feebleminded | defect | disorder | antisocial | aberration | phenotype | tuftsin | activation | gene | smad3 | dock2 |
| excremental | aerial | deficiency | deficiency | impaired | deformities | unstable | erythropoietic | abnormal | abnormal | glucosylation | bcl11b | atg5 | mitophagy |
| sybtem | defectivo | impaired | delinquents | adults | congenitally | lackof | masking | hexosephosphate | disorder | tuftsin | phagocyte | stat5b | repression |
| defectivo | overcrowding | congenitally | defectives | delinquents | mentally | impotence | defect | impairment | genetically | postreceptor | mosaic | orai1 | aberrant |
| storage | malformation | deterioration | imbeciles | incorrigibility | dentition | hyperactivity | deficiency | accelerated | sialyltransferase | peroxisomes | transactivating | opa1 | ikk2 |
| defeetive | defects | backwardness | moron | blind | crowding | conditioning | congenitally | metabolically | aberration | aberrant | downregulate | underexpression | gata3 |
| impairment | deterioration | abnormalities | delinquent | vision | malnutrition | germplasm | deficiencies | reutilized | depressed | galactosylation | organelles | upregulated | I κ Bα |

**Race**

| 1880–1889 | 1890–1899 | 1900–1909 | 1910–1919 | 1920–1929 | 1930–1939 | 1940–1949 | 1950–1959 | 1960–1969 | 1970–1979 | 1980–1989 | 1990–1999 | 2000–2009 | 2010–2019 |
|---|---|---|---|---|---|---|---|---|---|---|---|---|---|
| peoples | negro | jews | negro | happiness | jews | sex | sex | sex | sex | sex | ethnic | ethnic | ethnic |
| anglosaxon | beauty | offspring | extinction | peoples | religion | climate | marital | ethnic | ethnic | ethnic | sex | sex | sex |
| uncivilized | youth | youth | peoples | jews | negro | adolescent | nationality | religion | religion | age | ethnicity | hispanic | hispanic |
| savagery | ancestry | peoples | uncivilized | achievement | races | trait | ethnic | nationality | marital | socioeconomic | marital | ethnicity | ethnicity |
| offspring | social | intellectual | civilization | religion | populations | genetic | nativity | marital | nationality | urbanization | hispanic | nonblack | asian |
| religion | ape | barbarian | religion | habitat | dominant | environmental | religion | paternal | marriage | tadjusted | sociodemographic | marital | female |
| syphilized | humau | weaklings | weaklings | americans | patriarchal | religion | scx | spouses | habits | gender | nonblack | nonwhite | marital |
| debauched | natives | uncivilized | anglosaxon | youth | inherited | creed | distribution | status | socioeconomic | scx | parity | black | self |
| warlike | peoples | learning | innate | poverty | jewish | divorce | trait | nuns | nativity | ethnicity | socioeconomic | status | male |
| negro | sports | love | mating | intellectual | anglosaxon | status | porulation | partners | minorities | marital | nonwhite | nonhispanic | status |

Notes: All 200 most-related terms are available in Supplementary files.

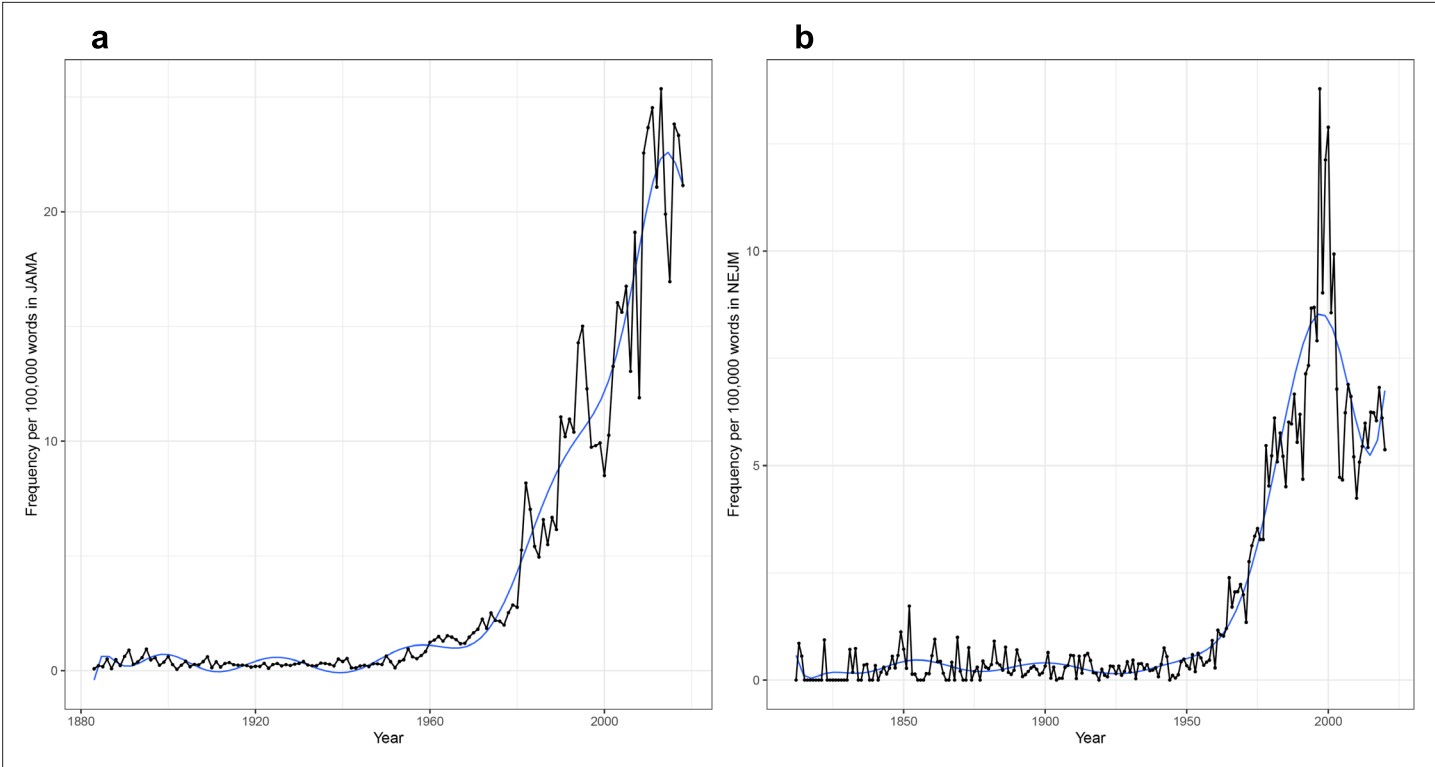

**Figure 6.** Time series plot, with frequency per 100,000 words as a function of year, for the word 'bias' in the (**a**) *Journal of the American Medical Association* (*JAMA*) corpus and the (**b**) *New England Journal of Medicine* (*NEJM*) corpus.

concerns with 'defective'. It might seem more benign to talk about defective genes than defective intellects and mental retardation, but the basic judgment remains: there is a normal standard against which variants are judged, and typically found to be inadequate, or 'defective'. This will inevitably contribute to the stigmatization of genetic variations associated with diseases or other undesirable traits. This is an interesting finding in the setting of the longstanding debate about whether the field of medical genetics is still colored by its eugenic prehistory (*Cowan, 2008*; *Comfort, 2012*).

Race has an even more striking history (*Tables 1 and 2*; *Supplementary files 9 and 10*). Its early occurrences in the 19th century cover enormous ground: sexual, debauched, monkey, peasanty, degradation, Jewish, anglosaxon, uncivilized, and many others. This should be no surprise. The racist aspects of medical theory and practice, from the 19th century through the 20th and into the 21st, have been well documented by historians (*Roberts, 2011*; *Wailoo, 2011*; *Braun, 2014*; *Vyas et al., 2020*). 'Jew' and 'Negro' are especially prominent in the eugenic decades of the early 20th century. A marked change occurs in the 1940s, when sex (and then ethnic/ethnicity) become race's closest associations, presumably a reflection of a shift in medical publication that linked sex and race in reporting the demographic characteristics of patients in clinical research trials (*Epstein, 2008*). This is not to say that racist bias had disappeared from medical theory. It is simply to note that a new use of race had emerged, a more administrative and bureaucratic one (e.g., the expectation that race, like age and gender, was a defining feature of patients that needed to be tracked in medical research and practice); as this usage became ubiquitous, it drowned out other uses and meanings of the word. By the 21st century, race had lost (at least in these word frequency associations) its early eccentric and eugenic associations and become an administrative category (ethnic, sex, Hispanic, Asian, female, marital, etc.). This routinization has many effects. It makes a normative judgment: doctors should think race – as a biological category – is relevant. And it obscures the ways in which race actually remained a contested category within medicine (*Tishkoff and Kidd, 2004*; *Pollock, 2012*; *Jones, 2021*). One clue concerning this discomfort is reflected by the emergence of 'self-identified' in *JAMA* after the 1990s: unsure of what race really is, doctors increasingly defaulted to putting responsibility on patients to 'self-identify' their race, as if that made the label somehow true or safe. However, as

many clinicians and researchers know, patients often struggle to shoehorn their complex identities into the limited options given to them (e.g., black versus nonblack, or the five races and two ethnic categories used widely in American health care). Once again, such a signal can lead to further examination through close textual analysis.

## Higher-level evolution of disease concepts

We can trace these histories in another way. The vector representations (or word addresses), from the related terms/word meaning analyses, can be clustered to discover higher-order structures (i.e., word groupings or neighborhoods). As described above, each word has a vector representation/word address based on the company it keeps. Two words that co-occur with high frequency will have similar vectors/addresses. Notably, for most words (e.g., 'the'), the vector/address is fixed and does not change with respect to time. This makes intuitive sense: the word 'the' has the same meaning in the 1810s as it does in the 2020s. However, for some words, the vector does vary and evolve over time. As noted in the word meaning analyses, this allows us to discover change in meaning as demonstrated by changing word contexts. For the word meanings analysis, this was limited to looking at a single word at any given time point.

Since the vectors/addresses of multiple words change over time, we can also trace the evolution of groupings or clusters of words; these groupings would represent 'higher-order' medical concepts. In simpler terms, rather than looking for related terms given a word of interest in a particular time period, we cluster all the words such that all related terms end up in a single group. Using a clustering technique called 'affinity propagation' that does not require us to pre-specify the number or the size of word neighborhoods/clusters (*Frey and Dueck, 2007*), we looked back through time (in quarter-century increments back to 1900) in *NEJM* to visualize the evolution of modern medical ideas in this abstract conceptual space (see Materials and methods). The advantage of this unsupervised clustering approach is that it is unaffected by our own subjectivity and pre-existing historical knowledge; that is, we do not a priori specify which words we are looking at or which groupings interest us.

The affinity propagation clustering algorithm generates an unspecified number of clusters every quarter century, starting from 1900. The clusters within each time period represent word groupings/neighborhoods, or equivalently, the algorithm's approximation to major conceptual clusters during that time. This is perhaps akin to how an alien civilization would understand disease concepts if they were to only read *NEJM* in that particular time period.

There were many clusters and stories to tell, but here we focus on one: the evolution in *NEJM* of our modern medical understanding of dementia (*Figure 7*). The algorithm identified a word cluster/neighborhood in the post-2000 time period, consisting of 14 terms: alzheimer, amyotrophic, angiopathy, dementia, frontotemporal, huntington, lewy, lobar, neurodegenerative, neuropathology, parkinson, pick, senile, and wernicke. These 14 terms had similar vectorrepresentations/word addresses in the post-2000 period, which we termed the 'dementia cluster'. In other words, these words tended to co-occur (i.e., be used) in similar contexts in *NEJM* articles published after the year 2000. Evidently, this current post-2000 dementia cluster is a narrow neurological one, which makes intuitive sense. We can subsequently trace each of these words back in time and explore which groupings each word previously inhabited (i.e., prior to the year 2000). The size of the connection between clusters denotes proportionally how many words from that cluster 'flow' into the next; the absence of a connection indicates that two consecutive clusters share no words. Moving backward like this, we can see a wider range of associations, with these 14 terms falling into different word neighborhoods/conceptual groups. This likely reflects several factors: increasing etiological and clinical characterization of mental disorders, changing burden of disease, changing language (e.g., the disappearance of senile dementia), and other issues (e.g., the 1950–1974 cluster on crime/law/malpractice has dissipated) (*Ballenger, 2006*).

This evolution of word neighborhoods is revealing in many ways. The 1900–1924 grouping of alcoholism, dementia, and neurosyphilis – as well as its persistence into the 1925–1949 period, reflects the fact that many of the pathological features associated with dementia were first observed in brains from individuals afflicted with general paresis (due to CNS syphilitic infection) and in individuals with chronic alcoholism (*Berchtold and Cotman, 1998*). In fact, Samuel Wilks first described atrophy (associated with decreased brain weight) as being due to chronic alcoholism and CNS syphilis ('neurosyphilis'), which was later investigated in relation to other dementias (*Berchtold and Cotman, 1998*).

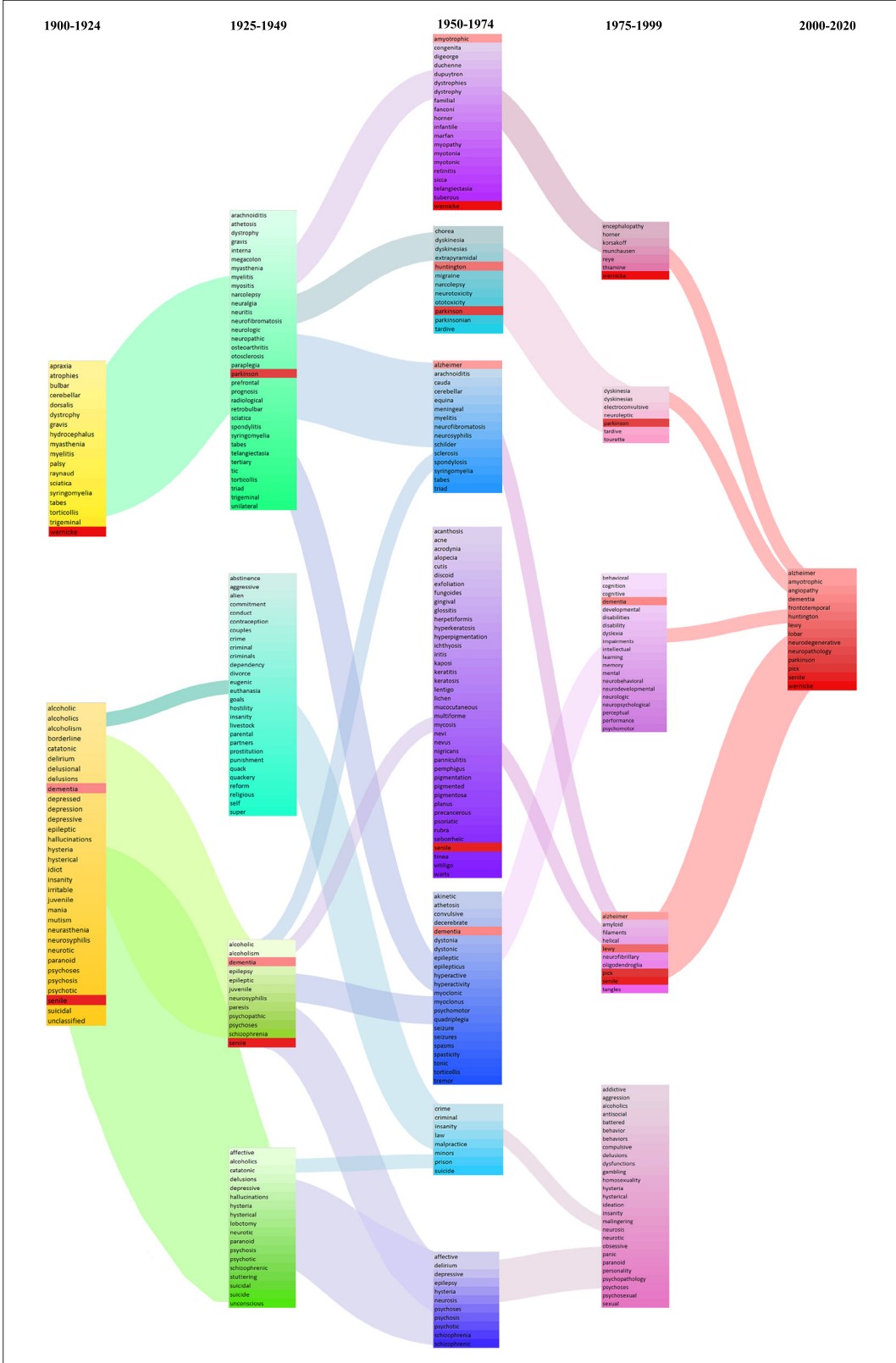

**Figure 7.** Evolution of the modern concept of 'dementia' as reflected in *New England Journal of Medicine* (*NEJM*). Each column denotes a quarter century: 1900–1924 (yellow), 1925–1949 (green), 1950–1974 (blue-purple), 1975–1999 (pink-violet), and 2000–2020 (red). Clustering using the word addresses/vectors generated during the 2000–2020 groups the following words together: alzheimer, amyotrophic, angiopathy, dementia, frontotemporal,

*Figure 7 continued on next page*

*Figure 7 continued*

huntington, lewy, lobar, neurodegenerative, neuropathology, parkinson, pick, senile, and wernicke (all highlighted with a red background). We subsequently identify which clusters, historically, each of these 14 terms belonged to. The size of the connections denotes proportionally how many words from that cluster 'flowed' into the next. The absence of a connection indicates that two consecutive clusters share no words.

We subsequently see the word groupings reflect the increased attention to the newly discovered pathological features – plaques and tangles – that had been identified in the brains of individuals with 'senile dementia'.

As with the other analyses, further research is required to distill and solidify our understanding of the modern conceptualization of dementia and Alzheimer's disease. Further, countless interrogations can be performed for the evolution of other modern medical concepts, and many of these will reveal unexpected signals and stories that demand further investigation.

## Conclusion

Tracing this evolution of medical knowledge and values, in the context of broader medical innovation and historical developments, exposes the shifting scientific, material, ethical, and epistemic frameworks of medicine over time. The trends and signals we have discovered have likewise impacted the ways patients and the public engage with medicine. Our analyses have only scratched the surface of what can be done with this novel methodology and database. Medicine has clearly changed in some ways that we expected. But there are many surprising things we did not expect (e.g., reduction in the mention of 'hospital' over the late 20th century, editorial differences in discussions surrounding abortion). While the signals we have uncovered require follow-up research using traditional methods of historical scholarship, further computational linguistic analyses of the content and language of publishing offer a treasure trove of insight waiting to be further explored. We invite others to join in this work.

## Materials and methods
### Constructing the *JAMA* and *NEJM* datasets

To enable this computational analysis, we constructed a database of all articles ever published in *JAMA* and *NEJM*. In this study, an article is defined as any document with a DOI, which is a persistent handle that can be used to identify academic publications uniquely.

For the *JAMA* dataset, we collected 304,905 DOIs, of which 223,748 had associated authorship metadata. Our *JAMA* database captured 278,461 articles published in *JAMA* from 1883 to 2018 (inclusive), representing >91% of all articles ever published. The missing articles (<9%) have not been digitized, have erroneous DOI references that could not be corrected, or simply, were not captured during crowd-sourcing efforts.

Similarly, based on DOI counts, our crowd-sourced *NEJM* database captured 182,675 articles published from 1812 to 2020, which represents >99.5% of all articles ever published in *NEJM*. The missing articles (estimated to be between 65 and 194 articles) may not have been captured in our dataset construction for several reasons. Some of these DOIs may be erroneous and/or do not resolve to a *NEJM* article (estimated to be 20 articles). Alternatively, the article may have been too recent or perhaps more simply, was not captured by our volunteers during our crowd-sourcing efforts. However, given that these articles represent less than 0.05–0.5% of the entire *NEJM* corpus and were missing at random, this would have little to no impact on subsequent analyses. For all articles, we curated article types and topics.

To validate the OCR results compared with the original manuscripts, two individuals manually and independently transcribed the same 600 lines randomly selected from each semi-century contained in our dataset (150 consecutive lines [composed of more than 31,000 characters] from each of the following time intervals: 1850–1899, 1900–1949, 1950–1999, 2000–2020). The two manually transcribed sets of lines were consolidated and manually verified by a third individual, with full access to the original manuscripts, to establish the gold standard with which we compared the automated OCR results with standard deviations bootstrapped from the data. We calculated the character error

rate to be (*Figure 8a*): 0.26% (± standard deviation [std] = 0.004%; 1 in 384 characters) for articles published before 1899, 0.014% (± std 0.001%; 1 in 7105 characters) for articles published between 1900 and 1949, 0.065% (± std 0.002%; 1 in 1520 characters) for articles published between 1950 and 1999, and 0.037% (± std 0.001%; 1 in 2701 characters) for articles published after 2000. All alphanumeric character discrepancies were traced back to typographical errors introduced by poor printing quality (e.g., e recognized as c when the [horizontal] bar of the letter e is missing; *Figure 8b*). This is further validated by the observation that 71% of character errors occurred in digitized manuscripts published prior to the year 1899, with a cumulative error rate of less than 1 in 2500 in the lines published after 1900. Notably, the error rate post-1900 was similar to the number of discrepancies between the two human transcribers. Post-2000, all OCR errors were traced back to difficulty in recognizing punctuation (e.g., long dashes) without a single error in alphanumeric recognition. During the manual transcription, we also noted the presence of typographical errors by the original authors (e.g., 'cnmmunicated' [sic] instead of 'communicated'; *Figure 8c*) and historical word choices ('essayed' [meaning tried], or 'farther investigation' [in lieu of 'further investigation']; *Figure 8d and e*). The automated OCR pipeline preserved these as did our human transcribers, which we believe is appropriate given the historical nature of the dataset.

## Shifting word meanings and evolution of disease concepts

As a broad overview, in less technical terms, for each decade (or quarter century), we calculated a 'vector representation' for all words in the *JAMA* and *NEJM* corpora separately. This 'vector representation' is the way the computer encodes the meaning of the word – with words that have similar meaning/used in similar contexts being correlated. As an analogy, we can encode location information as longitude and latitude coordinates. Places with similar coordinates would have similar climates. Similarly, words with correlated 'vectors' either have similar meanings or are used in similar contexts. The analysis described here can also be likened to 'guilty by association' – words that share similar company have similar 'vector representations'. For our shifting word meaning analyses, we find the top 200 most correlated words during that decade. Similarly, using these vector representations for each quarter century, we clustered the words into 'concepts', which we then manually traced to understand the evolution of broader disease and medical concepts (i.e., evolution of disease concept analyses).

Technical details: To analyze the evolution of word meaning, we used the modeling approach of temporal referencing (TR), detailed in *Dubossarsky et al., 2019*, which avoids post hoc alignment when learning individual word representations for different time periods. We performed the analysis separately for *JAMA* and *NEJM*. TR allows us to treat each journal's corpus as a single unit, rather than sub-dividing the articles into time-specific corpora. We a priori defined a crowd-sourced list of 10,497 words that we thought have historical and/or medical relevance (i.e., our target list). We then replaced each word from our target list with a time-specific token (either using decade intervals or quarter centuries, depending on the analysis). This allows our model to learn a vector for each word-time pair in a single space, and thus facilitates comparison without the need for alignment. The underlying model used to generate the word embeddings was a skim-gram with negative sampling architecture. For each word in our target list, to assess shifting word meanings, we used the cosine similarity metric to identify the top 200 most-correlated words for each decade. Sensitivity analyses were performed using a classical approach: sub-dividing the articles into time-specific corpora and generating embeddings limited to that time period. This classical approach did not require specification of a target list of words and supported the robustness of the TR approach. For the evolution of disease concepts, we used a longer time period (i.e., quarter century) to generate the word embeddings, which we subsequently clustered within each time period using the affinity propagation algorithm. Affinity propagation is a 'message-passing' clustering approach that, unlike k-means, does not require a priori specification of the number of clusters. To assess the robustness of the clusters and as further sensitivity analyses, we performed repeated running of affinity propagation (clustering) with different initialization parameters and seeds to validate the robustness of the clusters.

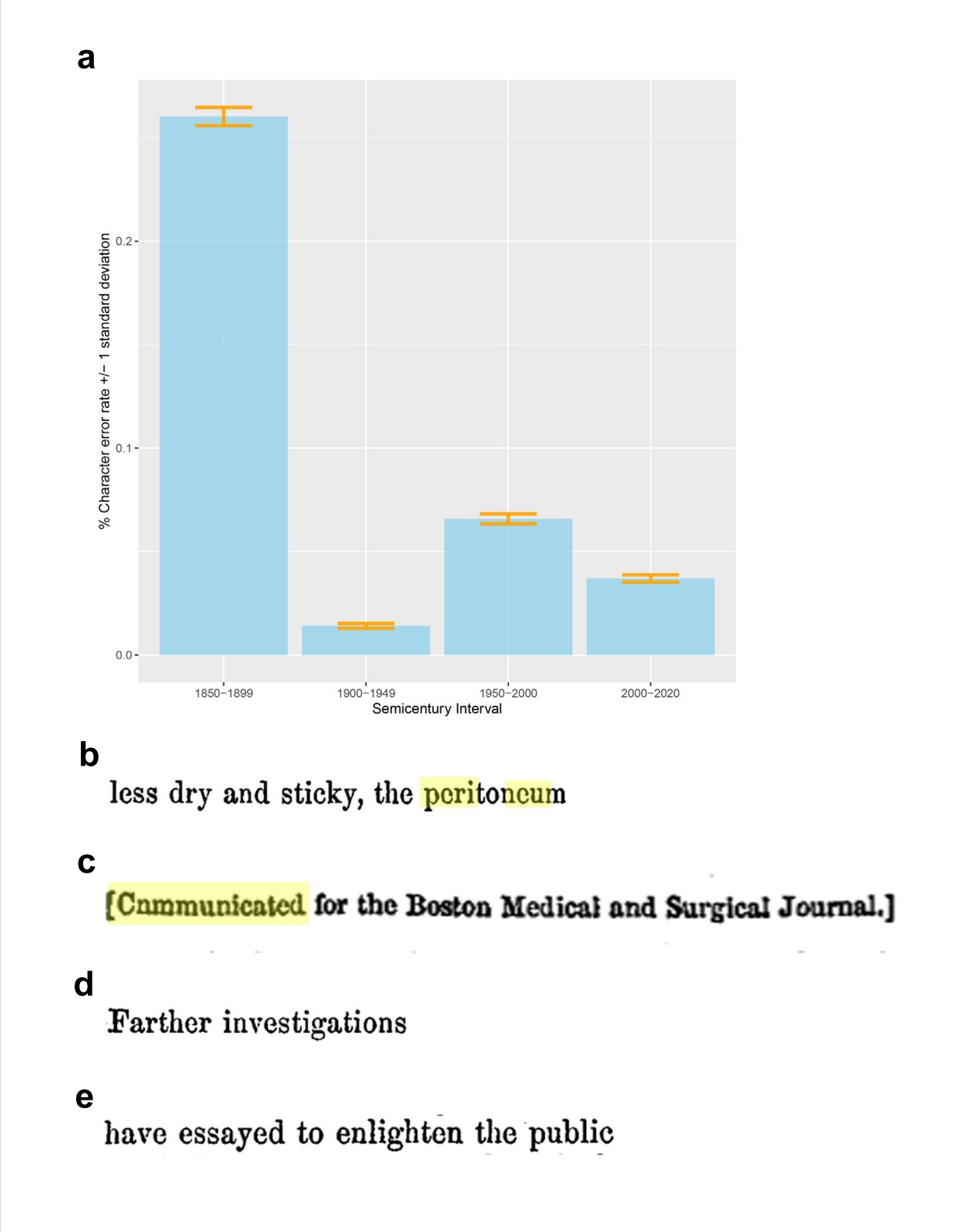

**Figure 8.** Validation and verification of optical character recognition (OCR) processing of the original *New England Journal of Medicine* (*NEJM*) and *Journal of the American Medical Association* (*JAMA*) manuscripts. (**a**) We estimated the character error rate, to be: 0.26% (± standard deviation [std] = 0.004%; 1 in 384 characters) for articles published before 1899, 0.014% (±std 0.001%; 1 in 7105 characters) for articles published between 1900 and 1949, 0.065% (±std 0.002%; 1 in 1520 characters) for articles published between 1950 and 1999, and 0.037% (±std 0.001%; 1 in 2701 characters) for articles

*Figure 8 continued*

published after 2000. (**b**) All alphanumeric character discrepancies were traced back to typographical errors introduced by poor printing quality (e.g., e recognized as c when the [horizontal] bar of the letter e is missing). During the manual transcription, we also noted the presence of: (**c**) typographical errors by the original authors and (**d, e**) historical word choices.

## Additional information

### Funding
No external funding was received for this work.

### Author contributions
Moustafa Abdalla, Conceptualization, Data curation, Formal analysis, Investigation, Methodology, Project administration, Software, Supervision, Validation, Visualization, Writing – original draft, Writing – review and editing; Mohamed Abdalla, Salwa Abdalla, Data curation, Formal analysis, Investigation, Methodology, Software, Validation, Visualization, Writing – review and editing; Mohamed Saad, Funding acquisition, Investigation, Methodology, Resources, Software, Supervision, Writing – review and editing; David S Jones, Formal analysis, Investigation, Methodology, Project administration, Validation, Visualization, Writing – original draft, Writing – review and editing; Scott H Podolsky, Conceptualization, Formal analysis, Investigation, Methodology, Project administration, Supervision, Validation, Visualization, Writing – original draft, Writing – review and editing

### Author ORCIDs
Moustafa Abdalla  http://orcid.org/0000-0002-2481-9753
David S Jones  http://orcid.org/0000-0003-0039-7784

### Decision letter and Author response
Decision letter https://doi.org/10.7554/eLife.72602.sa1
Author response https://doi.org/10.7554/eLife.72602.sa2

## Additional files

### Supplementary files
• Supplementary file 1. Table of the 200 most-related terms for the word 'patent' in *Journal of the American Medical Association* (*JAMA*), stratified per decade.

• Supplementary file 2. Table of the 200 most-related terms for the word 'patent' in *New England Journal of Medicine* (*NEJM*), stratified per decade.

• Supplementary file 3. Table of the 200 most-related terms for the word 'abortion' in *Journal of the American Medical Association* (*JAMA*), stratified per decade.

• Supplementary file 4. Table of the 200 most-related terms for the word 'abortion' in *New England Journal of Medicine* (*NEJM*), stratified per decade.

• Supplementary file 5. Table of the 200 most-related terms for the word 'bias' in *Journal of the American Medical Association* (*JAMA*), stratified per decade.

• Supplementary file 6. Table of the 200 most-related terms for the word 'bias' in *New England Journal of Medicine* (*NEJM*), stratified per decade.

• Supplementary file 7. Table of the 200 most-related terms for the word 'defective' in *Journal of the American Medical Association* (*JAMA*), stratified per decade.

• Supplementary file 8. Table of the 200 most-related terms for the word 'defective' in *New England Journal of Medicine* (*NEJM*), stratified per decade.

• Supplementary file 9. Table of the 200 most-related terms for the word 'race' in *Journal of the American Medical Association* (*JAMA*), stratified per decade.

• Supplementary file 10. Table of the 200 most-related terms for the word 'race' in *New England Journal of Medicine* (*NEJM*), stratified per decade.

• Transparent reporting form

## Data availability

Our manuscript analyses the full archives of NEJM and JAMA, which we are not permitted to share. Readers interested to access the original data must contact the editors of NEJM or JAMA to receive permission and the data. We are, however, able to share processed data, which we do in Supplementary file 1 to Supplementary file 10 and on our public ngram viewer (https://countway.harvard.edu/center-history-medicine/center-services/looking-glass). The data for the figures (e.g. spreadsheets used to generate the figure) may be downloaded from the ngram site or requested from the corresponding author.

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
