## [Editor Report]

This work analyzed more than half a million peer-reviewed articles published in two high-impact medical journals. It provides insights into the evolution of medical practice, language, and values over the past two centuries. Thus, it helps us contextualize our understanding of change in medicine and medical beliefs over time.

---

## [Decision Letter]

**Decision letter after peer review:**

Thank you for submitting your article "Through the Looking-Glass: Insights from Full Text Analyses of the Journal of the American Medical Association and the New England Journal of Medicine" for consideration by *eLife*. Your article has been reviewed by 2 peer reviewers, including Elham Mahmoudi as Reviewing Editor and Reviewer #1, and the evaluation has been overseen by Paul Noble as the Senior Editor. The following individual involved in review of your submission has agreed to reveal their identity: William J. Turkel (Reviewer #2).

Essential revisions:

The following is a list of essential revisions to improve the article:

1. Inevitable OCR errors may affect the results. Present evidence or describe methods used to validate OCR results compared with original manuscripts.

2. Word-based approaches inevitably lose the contextual information of surrounding words. The authors addressed this in the paper to some extent. The use of more sophisticated methods like phrasal units may improve the accuracy of the results.

3. Expand the methods section of the manuscript to address some of the missing methodological issues regarding data validation. For example, explain the methods used to ensure the accuracy of the data or describe the methods used to ensure the accuracy of the various text-mining techniques.

*Reviewer #1:*

This work is based on a preliminary exploration of a newly developed database of about a half million unique, published works in JAMA (1883-2018) and NEJM (1812-2020). Authors used crowd-sourcing and different innovative modeling techniques to explore changes in evolution of medical language, ethical considerations, practice of medicine, disease burden, and our understanding and perspectives of medical concepts over the past 200 years. The results offer an outlook into the ways patients, the public, and the medical community have interacted with medicine over time.

Strengths:

Constructing a digitized database of all unique articles published in JAMA and NEJM over the last 200 years is one of the main strengths of this work. Using open-source, optical character recognition (OCR) software, authors digitized the PDF version of all included manuscripts to extract critical information.

In their time-series analyses, authors went beyond the typical word count over time. Instead, they measured the proportion of the occurrences of objective words over total number of published words in every given year. This method (relative vs. absolute count over time) resulted in a more realistic outcome. It can be applied to future work, examining trends in use of other medical words.

To better decipher the temporal change in the meaning of specific terms, authors used a technique known as "word embedding." This technique uses a vector representation of a target word in any given time. This means that the vector captures not only the word but also neighboring words that occur more often with the targeted word. For some words, the vector representation does not change. For others, however, the vector evolves over time. Thus, the vector provides a temporal measure of change, if any, in the meaning of a targeted word.

Furthermore, the authors used the "affinity propagation" technique, which does not require determining the word neighborhood size in advance. This unsupervised, clustering approach is used to analyze change in grouping/clustering of words over time. The main strength of this approach is that it is not based on any prior assumption or knowledge. Thus, it is less prone to selection bias. The authors used this method to create several clusters in each given time. This method has many applications in research, allowing individuals to explore historical changes in various settings.

Weaknesses:

Although the development of a text-searchable, digitized database of this magnitude is a strength, the authors did not elaborate on what type of validation they performed to ensure accuracy (i.e., sensitivity and specificity) of the digitized database against actual manuscripts. Providing more information on data validation would add value to this manuscript and facilitate construction of similar data in the future.

Despite its advantages, interpretation of findings based on the unsupervised clustering method could be challenging. Our prior knowledge and accompanying conceptual frameworks that are built upon them would enable us to better interpret results.

In this study, authors used various text-mining or natural language processing techniques to analyze their constructed database. Although the manuscript is innovative and has numerous strengths, methods of validating the results have not been discussed. The sensitivity and specificity of the findings is unknown. The method section of the manuscript can be expanded to address some of the missing methodological issues. This would increase the likelihood of reusability of both data and methods.

It has been a pleasure reading this well-written and innovative work. Although neither the data nor the methods used are new, their application in analyzing the published work of the past 200 years is novel. The constructed, digitized database would be useful if authors are able and willing to make it available to other researchers. As the authors mentioned, they only scratch the surface with their research. This database can be used to analyze the main trends in disease burden and treatment approaches, etc.

I have two main suggestions to improve the presented work. It is not clear if any data or method validations were performed. Authors used an open OCR software to digitize about a half million PDF files. What method(s) have been used to ensure the accuracy of the data? Similarly, what methods have been used to ensure the accuracy of the various text-mining approaches? Did authors use any annotation technique to compare the results with the actual data (which is the gold standard)? Did they divide the data into training and test sets to measure accuracy, sensitivity, or specificity of their findings?

This manuscript is a great contribution to literature. Strengthening the method section would increase the impact of the work.

*Reviewer #2:*

Applying text mining techniques to almost half-a-million articles from the Journal of the American Medical Association (1883-2018) and the New England Journal of Medicine (1812-2020), the authors highlight a few of the ways that medical language has changed over the past two centuries. They explore changes in specific disease terms, words that signal the emergence of ethics and clinical trial infrastructure, and the dramatic rise (and puzzling decline after 1950) of the word 'hospital'. Using the technique of historical word embeddings, they also show semantic change in individual words over time. The word 'patent', for example, is primarily associated with patent medicines until about 1920, and with invention and congenital heart disease thereafter. They also explore the histories of the more culturally charged words 'abortion', 'bias', 'defective' and 'race'. Finally, they use an unsupervised clustering technique called affinity propagation to study the historical evolution of word clusters in semantic space, to trace 'higher-order' medical concepts.

This paper is clearly written and concise, practically a textbook exposition of basic methods of text mining and their considerable strengths.

It is, perhaps, a bit too concise, at least from the perspective of the historian. Each of the four substantive examples ('abortion', 'bias', etc.) could easily be developed into a much longer paper in its own right. That is a bit of a weakness.

A second weakness is that working with half-a-million articles in PDF form means that inevitable OCR errors affect the results to some extent. In the congenital heart disease example, the words 'ductus' and 'arteriosis' appear with 'patent' (as expected, since "patent ductus arteriosus" is a key phrase). The words 'duetus' and 'arteriosis' also appear very frequently, however. While the latter is a possible spelling mistake, the former is clearly a frequently-occurring OCR error. Anyone who has worked with OCR versions of late-19th-century or early-20th-century sources knows that this is typically a substantial source of error. Some measure of OCR quality might help evaluate the results.

A final weakness is that word-based approaches inevitably lose the contextual information of surrounding words. This is addressed in the paper to some extent by the use of unsupervised clustering on the word embeddings. Alternate approaches would be to use longer units, either linguistically naive ones (like n-grams) or more sophisticated ones (like phrasal units).

Despite these caveats, this paper is a very welcome addition to the literature, and accessible enough to be broadly useful.

Although the authors are not in a position to share their raw dataset, if they could make a pubicly accessible interface to the results of their analysis (as Google did with its Ngram Viewer) they could greatly increase the impact of their research.

---

## [Author Response]

Essential revisions:The following is a list of essential revisions to improve the article:1. Inevitable OCR errors may affect the results. Present evidence or describe methods used to validate OCR results compared with original manuscripts.

“To validate the OCR results compared with the original manuscripts, two individuals manually and independently transcribed the same 600 lines randomly selected from each semi-century contained in our dataset (150 consecutive lines [composed of more than 31,000 characters] from each of the following time intervals: 1850-1899, 1900-1949, 1950-1999, 2000-2020). The two manually transcribed sets of lines were consolidated and manually verified by a third individual, with full access to the original manuscripts, to establish the gold standard with which we compared the automated OCR results with standard deviations bootstrapped from the data. We calculated the character error rate to be (Figure 8a): 0.26% (± standard deviation [std] = 0.004%; 1 in 384 characters) for articles published before 1899, 0.014% (± std 0.001%; 1 in 7105 characters) for articles published between 1900 and 1949, 0.065% (± std 0.002%; 1 in 1520 characters) for articles published between 1950 and 1999, and 0.037% (± std 0.001%; 1 in 2701 characters) for articles published after 2000. All alphanumeric character discrepancies were traced back to typographical errors introduced by poor printing quality (e.g., e recognized as c when the [horizontal] bar of the letter e is missing; Figure 8b). This is further validated by the observation that 71% of character errors occurred in digitized manuscripts published prior to the year 1899, with a cumulative error rate of less than 1 in 2500 in the lines published after 1900. Notably, the error rate post-1900 was similar to the number of discrepancies between the two human transcribers. Post-2000, all OCR errors were traced back to difficulty in recognizing punctuation (e.g., long dashes) without a single error in alphanumeric recognition. During the manual transcription, we also noted the presence of typographical errors by the original authors (e.g., ‘cnmmunicated’[sic] instead of ‘communicated’; Figure 8c) and historical word choices (‘essayed’ [meaning tried], or ‘farther investigation’ [in lieu of ‘further investigation’]; Figure 8d, e). The automated OCR pipeline preserved these as did our human transcribers, which we believe is appropriate given the historical nature of the dataset.”

The above text and Figure 8 were added to our manuscript under the “Materials and methods” section.

2. Word-based approaches inevitably lose the contextual information of surrounding words. The authors addressed this in the paper to some extent. The use of more sophisticated methods like phrasal units may improve the accuracy of the results.

As suggested by Dr Turkel, we expanded our frequency analysis to include linguistically naïve n-grams (i.e., contiguous combinations of two or more words). We prefer linguistically naïve approaches to sophisticated ones (e.g., phrasal units) because they do not make any assumptions regarding meaning or usage of the phrase across the 200+ years contained in our analysis. Further, by using n-grams, we can discover ‘phrases’ limited to specific periods in time. Finally, as requested, we use the n-grams to provide an online free-to-access *NEJM* and *JAMA* n-gram viewer (similar to Google’s n-gram viewer), available at: https://countway.harvard.edu/center-history-medicine/center-services/looking-glass.

Figure 4 in the revised manuscript provides a high-level overview of the functionality available.

With regard to the n-gram analysis, we looked at the temporal frequency distribution for 10 bigrams (Figure 3): ‘personal equation’, ‘antibiotic resistance’, ‘informed consent’, ‘mental retardation’, ‘managed care’, ‘health policy’, ‘breast cancer’, ‘lung cancer’, ‘myocardial infarction’, and ‘heart attack’. The accompanying brief case studies included in our manuscript are also copied below:

“Analogous searches can be conducted for bigrams, compared to the denominator of all bigrams within a particular corpus (and they can be similarly conducted for still longer combinations of words). For example, this sentence contains six bigrams. As with single words, some of these examples confirm or lend nuance to expectations, methodologically or conceptually. Methodologically, it may be noted that one of us had previously manually traced the rise and fall of the term “personal equation” in the medical literature from 1850 to 1950 as a complex, multi-faceted term at times signifying individual patient or clinician variability, at other times signifying observer bias. The rise and fall of “personal equation” through the present bigram computation (Figure 3a) conforms well to the NEJM analysis conducted previously through examination of each instance of the term recognized by the NEJM’s own full-text search tool.^21”^

Turning to novel searches, we see that “antibiotic resistance” rises with the widespread recognition of resistant bacteria in the 1950s and of their horizontal transmission in the 1960s (Figure 3b, c). It appears to drop off in salience by the early 1980s, before a second wave of attention from the late 1980s onward as it became linked to emerging infections more broadly, and concerns about the capacity of the pharmaceutical industry to keep up with such newly resistant bacteria in particular.^22, 23^ The late 20th century rise of “informed consent” (apparently comprising half of the uses of “consent”; Figure 3d, e), fall of “mental retardation,” (Figure 3f, g) and swift rise and fall of “managed care” again (while “health policy” rose and remained high; Figure 3h, i) point to the ethical, semantic/conceptual, and material/infrastructural underpinnings of organized medicine.

Yet there are again perhaps more instructive signals. Both “breast cancer” and “lung cancer” rise throughout the latter half of the 20th century (Figure 3j, k). However, despite the predominance in recent decades of lung cancer mortality over breast cancer mortality, “breast cancer” is consistently mentioned at a higher rate (roughly twice the rate) than is “lung cancer.” This parallels discrepancies seen in the ratio of National Institutes of Health funding to the burden of disease contributed by these two forms of cancer. Does this represent differential stigmatization of lung cancer versus breast cancer patients? The mobilization of attention, advocacy, and funding around one versus the other condition? Again, such signals warrant additional investigation.

“Myocardial infarction,” like cancer, rises dramatically over the 20th century (Figure 3l, m). But here the trajectories in the two journals are distinct. While the plot of *NEJM* data shows a steady rise to its peak in 2000, the plot of *JAMA* data shows a bimodal distribution with one peak in the 1970s, followed by a notable dip, and then a rise to a second peak in the early 2000s. Similar trajectories are seen with “heart attack” in the two journals (though at much lower prevalence). It is hard to guess what might have led to the decrease in prevalence of myocardial infarction in JAMA at a time when it remained the leading cause of death. Did this reflect a shift in editorial policy? Was a different term briefly favored? Again, further investigation would be required.

To enable further research to distill these trends (both single word frequencies and higher-order ngrams), as well as to support other interrogations for unexpected signals and stories, we provide an ngram site to enable users to visualize and model longitudinal trends in *NEJM* and *JAMA* (Figure 4) at: https://countway.harvard.edu/center-history-medicine/center-services/looking-glass. 

3. Expand the methods section of the manuscript to address some of the missing methodological issues regarding data validation. For example, explain the methods used to ensure the accuracy of the data or describe the methods used to ensure the accuracy of the various text-mining techniques.

In addition to the details described under point 1 (which demonstrate the accuracy and validation of the OCR methods applied to the data), we have expanded the methods section to highlight: (a) our use of well-recognized and established trends in computational linguistics (e.g., the decrease of the word ‘the’) to support our frequency analysis; (b) our use of other independently published works/references (e.g. for the analysis of the bigram ‘personal equation’) that support the temporal trends that emerge from our dataset; (c) the independent generation of an ‘n-gram website’ to enable for verification and external validation of our figures; (d) sensitivity analyses performed using classical approaches (i.e. ‘classical’ word embeddings; sub-dividing the articles into time-specific corpora and generating embeddings limited to that time period) to support our modeling approach of Temporal Referencing (described in further detail in the text); and (e) repeated running of affinity propagation (clustering) with different initialization parameters and seeds to validate the robustness of the clusters.

Subsets of the above text, with additional details, were added in the appropriate locations under the methods section.

Reviewer #1:[…]Weaknesses:Although the development of a text-searchable, digitized database of this magnitude is a strength, the authors did not elaborate on what type of validation they performed to ensure accuracy (i.e., sensitivity and specificity) of the digitized database against actual manuscripts. Providing more information on data validation would add value to this manuscript and facilitate construction of similar data in the future.

We now highlight the extensive OCR validation and manual verification conducted (described in detail above). The validation methodology and accompanying results are also now incorporated in the revised “Materials and methods” section.

Despite its advantages, interpretation of findings based on the unsupervised clustering method could be challenging. Our prior knowledge and accompanying conceptual frameworks that are built upon them would enable us to better interpret results.

In addition to our previous analyses, we have expanded our supervised analyses to include ngrams, which leverage our prior knowledge and accompanying conceptual frameworks. We hope our dataset serves as both a support to prior knowledge and accompanying conceptual frameworks, as well as an unsupervised tool for hypothesis generation.

In this study, authors used various text-mining or natural language processing techniques to analyze their constructed database. Although the manuscript is innovative and has numerous strengths, methods of validating the results have not been discussed. The sensitivity and specificity of the findings is unknown. The method section of the manuscript can be expanded to address some of the missing methodological issues. This would increase the likelihood of reusability of both data and methods.

As discussed above and as incorporated in the now-revised methods section, we highlight both extensive OCR validation and several additional sensitivity analyses conducted to support the results of all analyses.

It has been a pleasure reading this well-written and innovative work. Although neither the data nor the methods used are new, their application in analyzing the published work of the past 200 years is novel. The constructed, digitized database would be useful if authors are able and willing to make it available to other researchers. As the authors mentioned, they only scratch the surface with their research. This database can be used to analyze the main trends in disease burden and treatment approaches, etc.I have two main suggestions to improve the presented work. It is not clear if any data or method validations were performed. Authors used an open OCR software to digitize about a half million PDF files. What method(s) have been used to ensure the accuracy of the data? Similarly, what methods have been used to ensure the accuracy of the various text-mining approaches? Did authors use any annotation technique to compare the results with the actual data (which is the gold standard)? Did they divide the data into training and test sets to measure accuracy, sensitivity, or specificity of their findings?This manuscript is a great contribution to literature. Strengthening the method section would increase the impact of the work.

As noted above, we’ve worked hard to strengthen the methods section by highlighting the extensive OCR validation and performing additional sensitivity analyses for all results.

Reviewer #2:Applying text mining techniques to almost half-a-million articles from the Journal of the American Medical Association (1883-2018) and the New England Journal of Medicine (1812-2020), the authors highlight a few of the ways that medical language has changed over the past two centuries. They explore changes in specific disease terms, words that signal the emergence of ethics and clinical trial infrastructure, and the dramatic rise (and puzzling decline after 1950) of the word 'hospital'. Using the technique of historical word embeddings, they also show semantic change in individual words over time. The word 'patent', for example, is primarily associated with patent medicines until about 1920, and with invention and congenital heart disease thereafter. They also explore the histories of the more culturally charged words 'abortion', 'bias', 'defective' and 'race'. Finally, they use an unsupervised clustering technique called affinity propagation to study the historical evolution of word clusters in semantic space, to trace 'higher-order' medical concepts.This paper is clearly written and concise, practically a textbook exposition of basic methods of text mining and their considerable strengths.It is, perhaps, a bit too concise, at least from the perspective of the historian. Each of the four substantive examples ('abortion', 'bias', etc.) could easily be developed into a much longer paper in its own right. That is a bit of a weakness.

We entirely agree that each of these examples could be developed into much longer papers. Indeed, our group includes historians who have collectively written on both experimental bias and concepts of race and racism in medicine. To address this weakness, we have nearly doubled the length of the aforementioned section (from 680 to 1100 words) and added a dozen new references. We would also like to state that our intention is that the tools demonstrated in this paper can provide data that may buttress (or refute or modify) existing hypotheses and narratives, or that may suggest novel signals to explore through more extended historical analyses using traditional methodologies of contextualization and evaluation.

A second weakness is that working with half-a-million articles in PDF form means that inevitable OCR errors affect the results to some extent. In the congenital heart disease example, the words 'ductus' and 'arteriosis' appear with 'patent' (as expected, since "patent ductus arteriosus" is a key phrase). The words 'duetus' and 'arteriosis' also appear very frequently, however. While the latter is a possible spelling mistake, the former is clearly a frequently-occurring OCR error. Anyone who has worked with OCR versions of late-19th-century or early-20th-century sources knows that this is typically a substantial source of error. Some measure of OCR quality might help evaluate the results.A final weakness is that word-based approaches inevitably lose the contextual information of surrounding words. This is addressed in the paper to some extent by the use of unsupervised clustering on the word embeddings. Alternate approaches would be to use longer units, either linguistically naive ones (like n-grams) or more sophisticated ones (like phrasal units).

As noted above, we have expanded our frequency analysis to include linguistically naïve n-grams. We looked at the temporal frequency distribution for 10 bigrams. The accompanying brief case studies included in our manuscript are copied above.

Despite these caveats, this paper is a very welcome addition to the literature, and accessible enough to be broadly useful.Although the authors are not in a position to share their raw dataset, if they could make a pubicly accessible interface to the results of their analysis (as Google did with its Ngram Viewer) they could greatly increase the impact of their research.

We use the ngram analyses described above to provide an online free-to-access *NEJM* and *JAMA* n-gram viewer (similar to Google’s n-gram viewer), available at: https://countway.harvard.edu/center-history-medicine/center-services/looking-glass.